# Efficient Convex Completion of Coupled Tensors using Coupled Nuclear Norms

Kishan Wimalawarne[1] and Hiroshi Mamitsuka[1,2]

[1]Bioinformatics Center, Kyoto University, Kyoto, Japan
[2]Department of Computer Science, Aalto University, Espoo, Finland
kishanwn@gmail.com, mami@kuicr.kyoto-u.ac.jp

## Abstract

Coupled norms have emerged as a convex method to solve coupled tensor completion. A limitation with coupled norms is that they only induce low-rankness using the multilinear rank of coupled tensors. In this paper, we introduce a new set of coupled norms known as coupled nuclear norms by constraining the CP rank of coupled tensors. We propose new coupled completion models using the coupled nuclear norms as regularizers, which can be optimized using computationally efficient optimization methods. We derive excess risk bounds for proposed coupled completion models and show that proposed norms lead to better performance. Through simulation and real-data experiments, we demonstrate that proposed norms achieve better performance for coupled completion compared to existing coupled norms.

## 1 Introduction

In this paper, we investigate convex coupled norms for coupled tensor completion. Two tensors are considered to be coupled when they share a common mode. A well explored problem with coupled tensors is coupled tensor completion, which studies imputation of partially observed tensors using coupled tensors as side information (Acar et al., 2014; Bouchard et al., 2013). Coupled tensor completion is commonly found in many real world applications such as link prediction (Ermis et al., 2015), recommendation systems (Acar et al., 2014) and computer vision (Li et al., 2015). Moreover, the increase in availability of data from multiple sources further makes coupled tensor completion an important research area requiring thorough investigation.

Over the years, several methods have been proposed to solve coupled tensor completion (Acar et al., 2014; Ermis et al., 2015). However, many of these methods are non-convex models leading to local optimal solutions. Additionally, these non-convex models have requirements of specifying ranks of coupled tensors, which are in many situations unknown. The recent development of coupled norms (Wimalawarne et al., 2018) has emerged as a convex solution for coupled completion. These coupled norms are modeled using the trace norm regularization, which eliminates the requirement of pre-specifying ranks. In spite of favorable qualities, coupled norms only induce low-rankness with respect to the multilinear rank of coupled tensors. This makes coupled norms sub-optimal for completion of coupled tensors with other low rank structures.

Until recently, most of the research on convex norms that induces low-rankness of tensors has focused on constraining the multilinear rank (Tomioka and Suzuki, 2013; Wimalawarne et al., 2014). However, recent studies (Yuan and Zhang, 2016) have shown that the tensor nuclear norm, which is a convex relaxation to minimizing the CANDECOMP/PARAFAC (CP) rank (Carroll and Chang, 1970; Harshman, 1970; Hitchcock, 1927; Kolda and Bader, 2009) has favorable properties compared low rank inducing norms that constrains the multilinear rank (Tomioka and Suzuki, 2013;

Wimalawarne et al., 2014). More specifically, Yang et al. (2015) showed that tensor completion using the tensor nuclear norm leads to better sample complexity compared to the overlapped norm (Tomioka and Suzuki, 2013; Liu et al., 2013), which was experimentally verified by Yuan and Zhang (2016). These advantages are unavailable for coupled norms since they do not support the tensor nuclear norm nor do they constrain the CP rank of coupled tensors.

In this paper, we investigate coupled completion through constraining the CP ranks of coupled tensors. We propose a set of convex coupled norms by extending the tensor nuclear norm. Additionally, we propose novel completion models that are regularized by the proposed norms, which obtain globally optimal solutions. We present theoretical analysis of the proposed completion models using excess risk bound analysis. Our analysis shows that the excess risk bound for two coupled $K$-mode tensors, $\mathcal{X} \in \mathbb{R}^{n \times \cdots \times n}$ and $Y \in \mathbb{R}^{n \times \cdots \times n}$, both having same CP rank $r$, is bounded by $O(r2^{4K}K\sqrt{n}(\ln n)^{K-1/2})$. We show that the obtained excess risk bounds are smaller compared to excess risk bounds resulting from multilinear rank based coupled norms. Finally, we verify our theoretical claims by simulation and real-data experiments.

We use the following notations throughout the paper. Given a $K$-mode tensor $\mathcal{T} \in \mathbb{R}^{n_1 \times \cdots \times n_K}$, we specify the mode-$k$ unfolding (Kolda and Bader, 2009) by $T_{(k)} \in \mathbb{R}^{n_k \times \prod_{j \neq k} n_j}$, which is obtained by concatenating all slices along the mode-$k$. Given two matrices, $M \in \mathbb{R}^{n_1 \times n_2}$ and $N \in \mathbb{R}^{n_1 \times n_2'}$, the notation $[M; N] \in \mathbb{R}^{n_1 \times (n_2 + n_2')}$ represents their concatenation on the common mode-1. We indicate the outer product between vectors $u_i \in \mathbb{R}^{n_i}$, $i = 1, \ldots, N$ using the notation $\otimes$ as $(u_1 \otimes \cdots \otimes u_N)_{i_1, \ldots, i_N} = \prod_{l=1}^{n} u_{l, i_l}$. The $k$-mode product of a tensor $\mathcal{T} \in \mathbb{R}^{n_1 \times \cdots \times n_k \cdots \times n_K}$ and a vector $v \in \mathbb{R}^{n_k}$ is defined as $\mathcal{T} \times_k v = \sum_{i_k=1}^{n_k} \mathcal{T}_{i_1, i_2, \ldots, i_k, \ldots, i_K} v_{i_k}$. Given that rank of the mode-$k$ unfolding of $\mathcal{T}$ is $r_k$, the multilinear rank of $\mathcal{T}$ is defined as $(r_1, \cdots, r_K)$.

## 2 Review of Coupled Completion

We briefly review existing coupled completion methods in this section.

### 2.1 Non-convex Factorization Methods

Coupled completion models have been mostly investigated through factorization methods. In essence, these methods consider explicit factorization of a coupled tensor $\mathcal{T} \in \mathbb{R}^{n_1 \times n_2 \times n_3}$ and a matrix $M \in \mathbb{R}^{n_1 \times m}$ as $\mathcal{T} = \sum_{i=1}^{R} a_i \otimes b_i \otimes c_i$ having $a_i \in \mathbb{R}^{n_1}$, $b_i \in \mathbb{R}^{n_2}, c_i \in \mathbb{R}^{n_3}$, $i = 1, \ldots, R$ and $M = \sum_{i=1}^{R} a_i \otimes d_i$ having $a_i \in \mathbb{R}^{n_1}$, $d_i \in \mathbb{R}^m$, $i = 1, \ldots, R$, respectively, with a common rank $R$ and shared components $a_i, i = 1, \ldots, R$. Many variations of factorization models for coupled completion models have been proposed based on CP decomposition with shared and unshared components (Acar et al., 2014), Tucker decomposition (Ermis et al., 2015), and non-negative factorization (Ermis et al., 2015). However, due to factorization, these coupled completion models are non-convex that lead to local optimal solutions. Furthermore, these methods require a priori specification of rank ($R$) of each tensor, as well as the number of shared components between the factorized tensors.

### 2.2 Convex Coupled Norms

Coupled norms (Wimalawarne et al., 2018) are a set of convex norms designed by combining low rank tensor and matrix norms. Given a tensor $\mathcal{T} \in \mathbb{R}^{n_1 \times n_2 \times n_3}$ and a matrix $M \in \mathbb{R}^{n_1 \times n_2'}$ coupled on mode $a$, coupled norms are defined as

$$\|\mathcal{T}, M\|_{(b,c,d)}^a,$$

where the subscripts $b, c, d \in \{O, L, S, -\}$ specify the regularization method to be applied to each mode and the superscript $a$ specifies the mode in which the tensor and the matrix are coupled. Notations O, L, and S indicate that the respective mode is regularized by using the overlapping trace norm (Tomioka and Suzuki, 2013), latent trace norm (Tomioka and Suzuki, 2013), and scaled latent trace norm (Wimalawarne et al., 2014), respectively, and $-$ indicates no regularization.

An example of a coupled norm that regularizes both coupled tensors using the overlapped trace norm is

$$\|\mathcal{T}, M\|_{(\mathrm{O,O,O})}^1 := \|[T_{(1)}; M]\|_{\mathrm{tr}} + \sum_{k=2}^3 \|T_{(k)}\|_{\mathrm{tr}}.$$

The following norm is another example where we consider the $\mathcal{T}$ as a summation of latent tensors $\mathcal{T}^{(1)}$, $\mathcal{T}^{(2)}$, and $\mathcal{T}^{(3)}$ and apply the scaled latent norm as

$$\|\mathcal{T}, M\|_{(\mathrm{S,S,S})}^1 = \inf_{\mathcal{T}^{(1)}+\mathcal{T}^{(2)}+\mathcal{T}^{(3)}=\mathcal{T}} \left( \frac{1}{\sqrt{n_1}} \|[T_{(1)}^{(1)}; M]\|_{\mathrm{tr}} + \sum_{k=2}^3 \frac{1}{\sqrt{n_k}} \|T_{(k)}^{(k)}\|_{\mathrm{tr}} \right).$$

Given two partially observed tensors $\hat{\mathcal{T}}_1 \in \mathbb{R}^{n_1 \times n_2 \times \cdots \times n_K}$, $K \geq 3$ and $\hat{\mathcal{T}}_2 \in \mathbb{R}^{n_1' \times n_2' \times \cdots \times n_{K'}'}$, $K' \geq 2$, coupled on their mode-$a$ with observed indexes given by the mappings $\Omega_{\hat{\mathcal{T}}_1}$ and $\Omega_{\hat{\mathcal{T}}_2}$, coupled completion is performed by solving

$$\min_{\mathcal{T}_1, \mathcal{T}_2} \frac{1}{2} \|\Omega_{\hat{\mathcal{T}}_1}(\mathcal{T}_1 - \hat{\mathcal{T}}_1)\|_{\mathrm{F}}^2 + \frac{1}{2} \|\Omega_{\hat{\mathcal{T}}_2}(\mathcal{T}_2 - \hat{\mathcal{T}}_2)\|_{\mathrm{F}}^2 + \lambda \|\mathcal{T}_1, \mathcal{T}_2\|_{\mathrm{cn}}^a,$$

where $\|\mathcal{T}_1, \mathcal{T}_2\|_{\mathrm{cn}}^a$ is a suitable coupled norm.

An important property with coupled norms is that the trace norm is applied with respect to each mode unfolding of tensors. This results in inducing low-rankness only by using the multilinear rank of coupled tensors. Furthermore, since the definitions of matrix rank and multilinear rank are different, concatenated regularization on the coupled mode may not be optimal for sharing information among the tensors.

## 3    Proposed Method: Coupled Completion via Coupled Nuclear Norms

In this section, we propose a set of convex coupled norms that overcome limitations of existing coupled completion methods. The main tool we use to build our norms is the tensor nuclear norm (Yuan and Zhang, 2016; Yang et al., 2015; Lim and Comon, 2014), which is defined for a tensor $\mathcal{T} \in \mathbb{R}^{n_1 \times n_2 \times \cdots \times n_K}$ as

$$\|\mathcal{T}\|_* = \inf \left\{ \sum_{j=1}^\infty \gamma_j \Big| \mathcal{T} = \sum_{j=1}^\infty \gamma_j u_{1j} \otimes u_{2j} \cdots \otimes u_{Kj}, \|u_{kj}\|_2^2 = 1, \gamma_j \geq \gamma_{j+1} > 0 \right\}. \quad (1)$$

In practice, we consider that $\mathcal{T}$ has a finite rank $R$, which is expressed by the notation $\mathrm{rank}(\mathcal{T}) = R$. When $K = 2$ and each $u_{kj}$ is orthogonal, the tensor nuclear norm is equivalent to the matrix nuclear norm.

We now propose coupled norms by only using the tensor nuclear norms, thus low-rankness of both the coupled tensors are induced using the CP rank. We name our norms *coupled nuclear norms*. We introduce the following notation to define the coupled nuclear norms for two coupled tensors $\mathcal{W} \in \mathbb{R}^{n_1 \times n_2 \times \cdots \times n_K}$ and $\mathcal{V} \in \mathbb{R}^{n_1' \times n_2' \times \cdots \times n_{K'}'}$ as

$$\|\mathcal{W}, \mathcal{V}\|_{\mathrm{ccp},(\lambda_\mathrm{b}, \mathrm{b})(\lambda_\mathrm{c}, \mathrm{c})}^a,$$

where the superscript $a$ indicates the coupled mode, and each tuple $(\lambda_b, b)$ and $(\lambda_c, c)$ indicates the regualarization method for each tensor. We specify $b, c \in \{\mathrm{F, L}\}$, where F and L indicate that a tensor is regularized as a whole or as a latent decomposition, respectively. Furthermore, we indicate $\lambda_b \in \mathbb{R}$ and $\lambda_c \in \mathbb{R}$ to specify regularization parameters for nuclear norms of each tensor. The subscript ccp is used to distinguish the proposed norms from coupled norms in (Wimalawarne et al., 2018).

Let us now look at a few definitions of coupled nuclear norms. We start with the following norm

$$\|\mathcal{W}, \mathcal{V}\|_{\mathrm{ccp},(\lambda_1,\mathrm{F})(\lambda_2,\mathrm{F})}^a = \left\{ \|\mathcal{W}\|_* \leq \lambda_1, \|\mathcal{V}\|_* \leq \lambda_2 \Big| \mathcal{W} = \sum_{i=1}^R \gamma_i x_{1i} \otimes \cdots \otimes x_{ai} \otimes \cdots x_{Ki}, \right.$$

$$\left. \mathcal{V} = \sum_{i=1}^R \nu_i y_{1i} \otimes \cdots \otimes x_{ai} \otimes \cdots y_{K'i} \right\}, \quad (2)$$

where the subscripts with F lead us to consider $\mathcal{W}$ and $\mathcal{V}$ as whole tensors without any latent decomposition. We assume that each tensor has a rank $R$ and all the component vectors $x_{ai},\ i = 1, \ldots, R$ on the coupled mode $a$ are common to both the tensors, while the tensor nuclear norm is applied to $\mathcal{W}$ and $\mathcal{V}$ to constrain their ranks.

A limitation in the previous norm is that it assumes both $\mathcal{W}$ and $\mathcal{V}$ have the same rank and all components along the coupled mode are common. In practice, this can be a strong assumption and we need to have more freedom for ranks and the amount of sharing among tensors. To incorporate these features into coupled nuclear norms, we propose to use latent decomposition of tensors, such that we learn latent tensors that are coupled to other tensors as well as uncoupled. Next, we assume a latent decomposition for $\mathcal{W}$ and define the following norm

$$\inf_{\mathcal{W}^{(1)}+\mathcal{W}^{(2)}=\mathcal{W}} \|\mathcal{W}, \mathcal{V}\|^a_{\mathrm{ccp},(\lambda_1,\lambda_2,\mathrm{L}),(\lambda_3,\mathrm{F})} = \inf_{\mathcal{W}^{(1)}+\mathcal{W}^{(2)}=\mathcal{W}} \left\{ \|\mathcal{W}^{(1)}\|_* \le \lambda_1, \|\mathcal{W}^{(2)}\|_* \le \lambda_2, \right.$$

$$\|\mathcal{V}\|_* \le \lambda_3 \left| \mathcal{W}^{(1)} = \sum_{i=1}^{R_1} \gamma_i^{(1)} x_{1i}^{(1)} \otimes \cdots \otimes x_{ai} \otimes \cdots x_{Ki}^{(1)}, \ \mathcal{W}^{(2)} = \sum_{i=1}^{R_2} \gamma_i^{(2)} x_{1i}^{(2)} \otimes \cdots \otimes x_{Ki}^{(2)}, \right.$$

$$\left. \mathcal{V} = \sum_{i=1}^{R_1} \nu_i y_{1i} \otimes \cdots \otimes x_{ai} \otimes \cdots y_{K'i} \right\}, \quad (3)$$

where the subscript $(\lambda_1, \lambda_2, \mathrm{L})$ indicates that the tensor $\mathcal{W}$ is considered as two latent tensors and their nuclear norms are constrained by $\lambda_1$ and $\lambda_2$. The third subscript $(\lambda_3, \mathrm{F})$ indicates that $\mathcal{V}$ is considered as a whole without any latent decomposition. Further, the norm considers $\mathcal{W}^{(1)}$ to have common factors with $\mathcal{V}$ due to coupling and $\mathcal{W}^{(2)}$ is independent from any coupling with $\mathcal{V}$. Due to the latent decomposition, the rank of $\mathcal{W}$ is $R_1 + R_2$, however, only $R_1$ components of $x_a$ in $\mathcal{W}$ are shared with $\mathcal{V}$.

In addition to the above coupled nuclear norms, we can further define $\inf_{\mathcal{V}^{(1)}+\mathcal{V}^{(2)}=\mathcal{V}} \|\mathcal{W}, \mathcal{V}\|^a_{\mathrm{ccp},(\lambda_1,\mathrm{F}),(\lambda_2,\lambda_3,\mathrm{L})}$ where the tensor $\mathcal{V}$ is considered to have a latent decomposition, and $\inf_{\mathcal{W}^{(1)}+\mathcal{W}^{(2)}=\mathcal{W},\mathcal{V}^{(1)}+\mathcal{V}^{(2)}=\mathcal{V}} \|\mathcal{W}, \mathcal{V}\|^a_{\mathrm{ccp},(\lambda_1,\lambda_2,\mathrm{L}),(\lambda_3,\lambda_4,\mathrm{L})}$ where both the tensors are considered to have latent decompositions. Furthermore, our proposed norms can be extended to define norms for coupled tensors with more than two coupled tensors.

It is important to note that the definition of proposed norms do not adhere to all the properties of the normed space. Rather, they can be considered them as sets constructed by tensor nuclear norms. However, we refer to our definitions as norms since they are constructed by constraining the tensor nuclear norms. Further, we point out that the number of different norms we need for a coupled tensor using coupled nuclear norms are less compared to multilinear rank based coupled norms (Wimalawarne et al., 2018).

### 3.1   New Coupled Completion Models

We now propose coupled completion models using coupled nuclear norms. Let us consider two partially observed tensors $\mathcal{X} \in \mathbb{R}^{n_1 \times n_2 \times \cdots \times n_K}$ and $\mathcal{Y} \in \mathbb{R}^{n'_1 \times n'_2 \times \cdots \times n'_{K'}}$ coupled on the mode $a$. Let us also consider $\Omega_1 : \mathbb{R}^{n_1 \times n_2 \times \cdots \times n_K} \to \mathbb{R}^{m_1}$ and $\Omega_2 : \mathbb{R}^{n'_1 \times n'_2 \times \cdots \times n'_{K'}} \to \mathbb{R}^{m_2}$ as mapping to observed elements of $\mathcal{X}$ and $\mathcal{Y}$, respectively, where $m_1$ and $m_2$ are the number of observed elements.

Our objective is to impute missing elements of $\mathcal{X}$ and $\mathcal{Y}$ by performing coupled completion using our proposed norms. Let $\mathcal{W}$ and $\mathcal{V}$ be completed tensors that we want to obtain for $\mathcal{X}$ and $\mathcal{Y}$, respectively. To achieve this using $\|\mathcal{W}, \mathcal{V}\|^a_{\mathrm{ccp},(\lambda_1,\mathrm{F})(\lambda_1,\mathrm{F})}$, we propose a completion model as

$$\min_{\mathcal{W},\mathcal{V}} \|\mathcal{W}, \mathcal{V}\|^a_{\mathrm{ccp},(\lambda_1,\mathrm{F})(\lambda_1,\mathrm{F})}$$

$$\text{s.t } \Omega_1(\mathcal{W}) = \Omega_1(\mathcal{X}), \quad \Omega_2(\mathcal{V}) = \Omega_2(\mathcal{Y}), \quad (4)$$

and another completion model by using $\|\mathcal{W}, \mathcal{V}\|^a_{\mathrm{ccp},(\lambda_1,\lambda_2,\mathrm{L})(\lambda_3,\mathrm{F})}$ as

$$\min_{\mathcal{W}^{(1)}+\mathcal{W}^{(2)}=\mathcal{W},\mathcal{V}} \|\mathcal{W}, \mathcal{V}\|^a_{\mathrm{ccp},(\lambda_1,\lambda_2,\mathrm{L})(\lambda_1,\mathrm{F})}$$

$$\text{s.t } \Omega_1(\mathcal{W}^{(1)} + \mathcal{W}^{(2)}) = \Omega_1(\mathcal{X}),$$

$$\Omega_2(\mathcal{V}) = \Omega_2(\mathcal{Y}). \quad (5)$$

Similarly, we can define completion models using $\inf_{\mathcal{V}^{(1)}+\mathcal{V}^{(2)}=\mathcal{V}} \|\mathcal{W},\mathcal{V}\|^a_{\mathrm{ccp},(\lambda_1,\mathrm{F}),(\lambda_2,\lambda_3,\mathrm{L})}$ and $\inf_{\mathcal{W}^{(1)}+\mathcal{W}^{(2)}=\mathcal{W},\mathcal{V}^{(1)}+\mathcal{V}^{(2)}=\mathcal{V}} \|\mathcal{W},\mathcal{V}\|^a_{\mathrm{ccp},(\lambda_1,\lambda_2,\mathrm{L}),(\lambda_3,\lambda_4,\mathrm{L})}$.

A key advantage with the proposed coupled nuclear norms is that they do not have overlapping group structures as in (Wimalawarne et al., 2018) and all tensors are regularized separately. This allows us to use a computationally feasible method such as the Frank-Wolfe optimization (Jaggi, 2013) to solve the proposed completion models. We provide a Frank-Wolfe based optimization method to solve above completion models in the Section B of the Appendix.

## 4 Theoretical Analysis

In this section, we analyze excess risk bounds for proposed coupled completion using coupled nuclear norms. We consider a partially observed $K$-mode tensor $\mathcal{X} \in \mathbb{R}^{n \times \cdots \times n}$ and a partially observed $K'$-mode tensor $\mathcal{Y} \in \mathbb{R}^{n \times \cdots \times n}$ coupled on their first modes. Let us consider two sets S and P, whose elements contain indexes of arbitrary subsets of elements of $\mathcal{X}$ and $\mathcal{Y}$, respectively. Following (Shamir and Shalev-Shwartz, 2014), we split S and P uniformly at random into training and test sets; the set S as $\mathrm{S_{Train}}$ and $\mathrm{S_{Test}}$ such that $\mathrm{S} = \mathrm{S_{Ttrain}} \cup \mathrm{S_{Test}}$, the set P as $\mathrm{P_{Train}}$ and $\mathrm{P_{Test}}$ such that $\mathrm{P} = \mathrm{P_{Ttrain}} \cup \mathrm{P_{Test}}$. Furthermore, following (Shamir and Shalev-Shwartz, 2014) we consider the special case where $|\mathrm{S_{Train}}| = |\mathrm{S_{Test}}| = |\mathrm{S}|/2$ and $|\mathrm{P_{Train}}| = |\mathrm{P_{Test}}| = |\mathrm{P}|/2$.

To prove excess risk bounds, we recast each coupled nuclear norm as a hypothesis class for each completion model. Let us again denote $\mathcal{W}$ and $\mathcal{V}$ as completed tensors we want to learn from $\mathcal{X}$ and $\mathcal{Y}$, respectively. Given the coupled nuclear norm $\|\mathcal{W},\mathcal{V}\|^a_{\mathrm{ccp},(\lambda_b,\mathrm{F})(\lambda_c,\mathrm{F})}$, we define a hypothesis class as $\mathsf{W} = \{\mathcal{W},\mathcal{V} : \|\mathcal{W},\mathcal{V}\|^a_{\mathrm{ccp},(\lambda_b,\mathrm{F})(\lambda_c,\mathrm{F})}, \mathrm{rank}(\mathcal{W}) = \mathrm{rank}(\mathcal{V}) = r\}$ for some regularization parameter $\lambda_b$ and $\lambda_c$. Using the hypothesis class and a $\Lambda$-Lipschitz continuous and $b_l$ bounded loss function $l(\cdot,\cdot)$ that measures the difference between the predicted and actual values, we write the average loss over training sets of a coupled completion model as

$$L_{\mathrm{S_{Train}},\mathrm{P_{Train}}}(\mathcal{W},\mathcal{V}) := \frac{1}{|\mathrm{S_{Train}} \cup \mathrm{P_{Train}}|}\left[ \sum_{(i_1,\ldots,i_K) \in \mathrm{S_{Train}}} l(\mathcal{X}_{i_1,\ldots,i_K},\mathcal{W}_{i_1,\ldots,i_K}) \right.$$
$$\left. + \sum_{(j_1,\ldots,j_{K'}) \in \mathrm{P_{Train}}} l(\mathcal{Y}_{j_1,\ldots,j_{K'}},\mathcal{V}_{j_1,\ldots,j_{K'}}) \right], \quad (6)$$

and the average loss over test sets can be constructed similarly as $L_{\mathrm{S_{Test}},\mathrm{P_{Test}}}(\mathcal{W},\mathcal{V})$ by substituting $\mathrm{S_{Train}}$ and $\mathrm{P_{Train}}$ in (6) with $\mathrm{S_{Test}}$ and $\mathrm{P_{Test}}$, respectively.

By the transductive Rademacher complexity theory (El-Yaniv and Pechyony, 2007; Shamir and Shalev-Shwartz, 2014), the excess risk can be upper bounded with the probability $1 - \delta$ as

$$L_{\mathrm{S_{Test}},\mathrm{P_{Test}}}(\mathcal{W},\mathcal{V}) - L_{\mathrm{S_{Train}},\mathrm{P_{Train}}}(\mathcal{W},\mathcal{V}) \leq 4R_{\mathrm{S,P}}(l \circ \mathcal{W}, l \circ \mathcal{V}) + b_l\left( \frac{11 + 4\sqrt{\log \frac{1}{\delta}}}{\sqrt{|\mathrm{S_{Train}} \cup \mathrm{P_{Train}}|}} \right), \quad (7)$$

where $R_{\mathrm{S,P}}(l \circ \mathcal{W}, l \circ \mathcal{V})$, which is expressed as

$$R_{\mathrm{S,P}}(l \circ \mathcal{W}, l \circ \mathcal{V}) = \frac{1}{|\mathrm{S} \cup \mathrm{P}|}\mathbb{E}_\sigma\left[ \sup_{\mathcal{W},\mathcal{V} \in \mathsf{W}} \sum_{i_1,\ldots,i_K} \Sigma_{i_1,\ldots,i_K} l(\mathcal{X}_{i_1,\ldots,i_K},\mathcal{W}_{i_1,\ldots,i_K}) + \right.$$
$$\left. \sum_{j_1,\ldots,j_{K'}} \Sigma'_{j_1,\ldots,j_{K'}} l(\mathcal{Y}_{j_1,\ldots,j_{K'}},\mathcal{V}_{j_1,\ldots,j_{K'}}) \right], \quad (8)$$

where $\Sigma \in \mathbb{R}^{n \times n \times \cdots n}$ and $\Sigma' \in \mathbb{R}^{n \times n \times \cdots n}$ are $K$-mode and $K'$-mode tensors, respectively, and $\Sigma_{i_1,\ldots,i_K} = \sigma_\phi \in \{-1,1\}$ with probability 0.5 if $(i_1,..,i_K) \in \mathrm{S}$ belonging to an index $\phi \in 1,\ldots,|\mathrm{S}|$ or $\Sigma_{i_1,\ldots,i_K} = 0$ otherwise, and $\Sigma'_{j_1,\ldots,j_{K'}} = \sigma_{|\mathrm{S}|+\phi'} \in \{-1,1\}$ with probability 0.5 if $(j_1,..,j_{K'}) \in \mathrm{P}$ belonging to an index $\phi' \in 1,\ldots,|\mathrm{P}|$ or $\Sigma'_{j_1,\ldots,j_{K'}} = 0$ otherwise.

In the next two theorems, we show the Rademacher complexities for proposed coupled nuclear norms $\|\mathcal{W},\mathcal{V}\|_{\mathrm{ccp},(\lambda_1,\mathrm{F}),(\lambda_1,\mathrm{F})}$ and $\|\mathcal{W},\mathcal{V}\|_{\mathrm{ccp},(\lambda_1,\lambda_2,\mathrm{L}),(\lambda_3,\mathrm{F})}$ (Detailed proof of these theorems are given in Section C of appendix.)

**Theorem 1.** *Let us consider* $\|\mathcal{W},\mathcal{V}\|^a_{\mathrm{ccp},(\lambda_1,\mathrm{F})(\lambda_2,\mathrm{F})}$ *and its associated hypothesis class as* $\mathsf{W} = \{\mathcal{W},\mathcal{V} : \|\mathcal{W},\mathcal{V}\|^a_{\mathrm{ccp},(\lambda_1,\mathrm{F})(\lambda_1,\mathrm{F})}, \mathrm{rank}(\mathcal{W}) = \mathrm{rank}(\mathcal{V}) = r\}$. *Then the Rademacher complexity is bounded as*

$$R_{\mathrm{S},\mathrm{P}}(l \circ \mathcal{W}, l \circ \mathcal{V}) \leq \frac{c\Lambda}{|\mathrm{S} \cup \mathrm{P}|}\Bigg[ rB_{\mathcal{W}}2^{3K+K'}K\sqrt{n}(\ln n)^{K-1/2}$$
$$+ rB_{\mathcal{V}}2^{K+3K'}K'\sqrt{n}(\ln n)^{K'-1/2}\Bigg].$$

*where* $\gamma_1 \leq B_{\mathcal{W}}$ *and* $\nu_1 \leq B_{\mathcal{V}}$ *of* (2) *and* $c$ *is a constant.*

**Theorem 2.** *Let us consider* $\|\mathcal{W},\mathcal{V}\|^a_{\mathrm{ccp},(\lambda_1,\lambda_2,\mathrm{L})(\lambda_3,\mathrm{F})}$ *and its hypothesis class as* $\mathsf{W} = \{\mathcal{W}^{(1)}, \mathcal{W}^{(2)}, \mathcal{V} : \inf_{\mathcal{W}^{(1)}+\mathcal{W}^{(2)}=\mathcal{W}} \|\mathcal{W},\mathcal{V}\|^a_{\mathrm{ccp},(\lambda_1,\lambda_2,\mathrm{L})(\lambda_3,\mathrm{F})}, \mathrm{rank}(\mathcal{W}^{(1)}) = \mathrm{rank}(\mathcal{V}) = r_1, \mathrm{rank}(\mathcal{W}^{(2)}) = r_2\}$. *Then the Rademacher complexity is bounded as*

$$R_{\mathrm{S},\mathrm{P}}(l \circ \mathcal{W}, l \circ \mathcal{V}) \leq \frac{c\Lambda}{|\mathrm{S} \cup \mathrm{P}|}\Bigg[ (r_1 B_{\mathcal{W}_1} + r_2 B_{\mathcal{W}_2})2^{3K+K'}K\sqrt{n}(\ln n)^{K-1/2}$$
$$+ r_2 B_{\mathcal{V}}2^{K+3K'}K'\sqrt{n}(\ln n)^{K'-1/2}\Bigg].$$

*where* $\gamma_1^{(1)} \leq B_{\mathcal{W}_1}$, $\gamma_1^{(2)} \leq B_{\mathcal{W}_2}$, $\nu_1 \leq B_{\mathcal{V}}$ *of* (3) *and* $c$ *is a constant.*

The Rademacher complexities in Theorems 1 and 2 show that for $K \geq K'$ and $r = r_1 = r_2$, excess risks is bounded by $\mathrm{O}(r2^{4K}K\sqrt{n}(\ln n)^{K-1/2})$. The excess risk bound for the coupled norm in (Wimalawarne et al., 2018) for the coupled tensors with multilinear rank $(r', \ldots, r')$ are bounded by $\mathrm{O}(\sqrt{r'}K[\sqrt{n^{K-1}} + \sqrt{n}])$. Since coupled nuclear norms are bounded by $\sqrt{n}(\ln n)^{K-1/2}$ compared to coupled norms (Wimalawarne et al., 2018) that are bounded by $\sqrt{n^{K-1}}$, coupled nuclear norms can lead to lower excess risk when coupled tensors have large dimensions ($n$ and $K$ are large). Though CP rank and multilinear rank cannot be compared directly, when the CP rank is smaller than the mode dimensions ($r < n$) our theoretical analysis shows that coupled nuclear norms are more capable of better performance compared to multilinear rank based coupled norms. Additionally, the Rademacher complexity is divided by the total number of observed samples from both the coupled tensors ($|\mathrm{S} \cup \mathrm{P}|$) leading to a lower Rademacher complexity compared to separate tensor completions.

## 5 Experiments

We carried out several simulations and real world data experiments to evaluate empirical performances of our proposed methods.

### 5.1 Simulation Experiments

We designed simulation experiments using coupled tensors using both the CP rank and the multilinear rank. For each simulation, we created a tensor $\mathcal{T} \in \mathbb{R}^{20 \times 20 \times 20}$ and a matrix $M \in \mathbb{R}^{20 \times 30}$ with specified ranks and coupled them on their first modes (without losing generality) by sharing a certain amount of singular components along the first mode. We chose these dimensions of coupled tensors in accordance with simulation experiments in (Wimalawarne et al., 2018) for easier comparison. In order to create a tensor $\mathcal{T}$ with CP rank of $r$, we generated the tensor as $\mathcal{T} = \sum_{a=1}^{r} \zeta_a u_a \otimes v_a \otimes w_a$ where $u_a \in \mathbb{R}^{20}$, $v_a \in \mathbb{R}^{20}$, and $w_a \in \mathbb{R}^{20}$ are randomly generated unit vectors and $\zeta_a \in \mathbb{R}$. To create $\mathcal{T}$ with multilinear rank of $(r_1, r_2, r_3)$, we generated orthogonal matrices $U \in \mathbb{R}^{20 \times r_1}$, $V \in \mathbb{R}^{20 \times r_2}$, and $W \in \mathbb{R}^{20 \times r_3}$, and a core tensor $\mathcal{C} \in \mathbb{R}^{r_1 \times r_2 \times r_3}$ with elements randomly sampled from a Normal distribution and compute $\mathcal{T} = \mathcal{C} \times_1 U \times_2 V \times_3 W$. We also created a rank $R$ matrix $M = XSY^\top$ with orthogonal matrices $X \in \mathbb{R}^{20 \times R}$ and $Y \in \mathbb{R}^{R \times 30}$, and a diagonal matrix $S \in \mathbb{R}^{R \times R}$ randomly generated from $\mathbb{R}^+$. We coupled the $\mathcal{T}$ and $M$ by sharing $r'$ components between them as $X(:, 1 : r') = [u_1, \ldots, u_{r'}]$ for CP rank based tensors and $X(:, 1 : r') = U(:, 1 : r')$ for multilinear rank based tensors. In order to generate datasets for simulations, we selected training sets of 30, 50, and 70 percentages from total number of elements of the tensor and the matrix, 10 percent as validation sets and the rest as test sets. For each simulation we repeated experiments with 10 random selections.

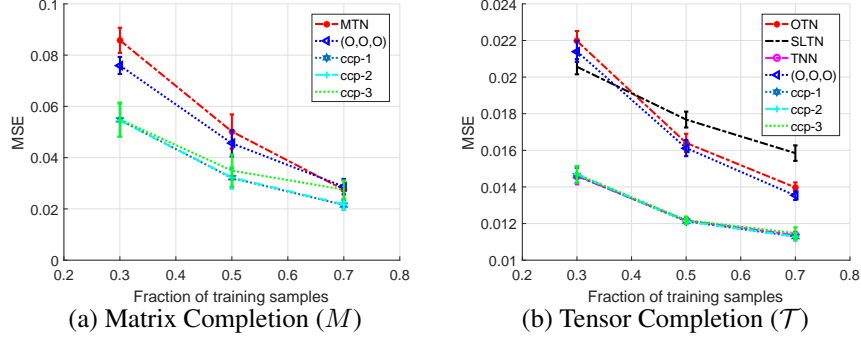

(a) Matrix Completion ($M$)    (b) Tensor Completion ($\mathcal{T}$)

Figure 1: Performances of completion of the tensor with dimensions of $20 \times 20 \times 20$ and CP rank of 5 and matrix with dimensions of $20 \times 30$ and rank of 5 both sharing 5 components.

We experimented with proposed coupled nuclear norms [1] $\|\mathcal{T}, M\|_{\mathrm{ccp},(\lambda_1,\mathrm{F}),(\lambda_1,\mathrm{F})}$, $\|\mathcal{T}, M\|_{\mathrm{ccp},(\lambda_2,\lambda_3,\mathrm{L}),(\lambda_2,\mathrm{F})}$, and $\|\mathcal{T}, M\|_{\mathrm{ccp},(\lambda_4,\mathrm{F}),(\lambda_4,\lambda_5,\mathrm{L})}$. For visual convenience in figures, we use shortened names for $\|\mathcal{T}, M\|_{\mathrm{ccp},(\lambda_1,\mathrm{F}),(\lambda_1,\mathrm{F})}$, $\|\mathcal{T}, M\|_{\mathrm{ccp},(\lambda_2,\lambda_3,\mathrm{L}),(\lambda_2,\mathrm{F})}$, and $\|\mathcal{T}, M\|_{\mathrm{ccp},(\lambda_4,\mathrm{F}),(\lambda_4,\lambda_5,\mathrm{L})}$ as ccp-1, ccp-2, and ccp-3, respectively. For all these norms, we used the regularization parameters $\lambda_1, \ldots, \lambda_5$ in the range from 0.01 to 50 with intervals of 1. As baseline methods, we performed completion of each individual tensor using the overlapped trace norm (OTN) and the scaled latent trace norm (SLTN) and individual matrix completion using the matrix trace norm (MTN). We also used the tensor nuclear norm as a baseline method to evaluate individual tensor completion. Additionally, we performed coupled completion with coupled norms (Wimalawarne et al., 2018). However, due to the difficulty in plotting all the norms in a single graph only the result from the best coupled norm is plotted. For all the baseline methods, we selected the optimal regularization parameters from the range of 0.01 to 5 in divisions of 0.025.

For our first experiment we created $\mathcal{T}$ by specifying a CP rank of 5 and $M$ with rank of 5. We coupled $\mathcal{T}$ and $M$ by sharing all components on their first modes. Figure 1 shows that coupled nuclear norms have outperformed individual completion of the tensor and the matrix, as well as coupled completion by the coupled norm $(\mathrm{O}, \mathrm{O}, \mathrm{O})$.

Next, we give a simulation experiment with coupled tensor using multilinear ranks. We constructed $\mathcal{T}$ with multilinear rank of $(5, 5, 5)$ and $M$ with rank of 5 and shared all components on the first mode. Figure 2 shows that the proposed coupled nuclear norms $\|\mathcal{T}, M\|_{\mathrm{ccp},(\lambda_1,\mathrm{F}),(\lambda_1,\mathrm{F})}$ and $\|\mathcal{T}, M\|_{\mathrm{ccp},(\lambda_2,\lambda_3,\mathrm{L}),(\lambda_2,\mathrm{F})}$ have outperformed $(\mathrm{O}, \mathrm{O}, \mathrm{O})$ for both tensor and matrix completion indicating that proposed norms are versatile for coupled tensors with multilinear ranks.

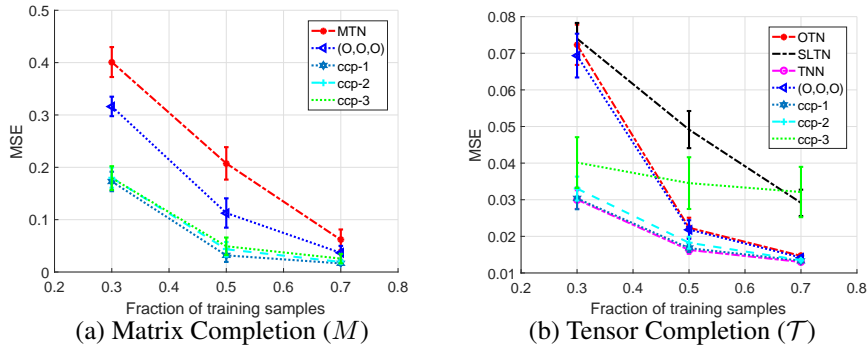

(a) Matrix Completion ($M$)    (b) Tensor Completion ($\mathcal{T}$)

Figure 2: Performances of completion of the tensor with dimensions of $20 \times 20 \times 20$ and multilinear rank of $(5, 5, 5)$ and matrix with dimensions of $20 \times 30$ and rank of 5 both sharing 5 components.

In all the above experiments, coupled nuclear norms have performed comparable or better than individual tensor and matrix completion. We give further simulation experiments in Section D of the Appendix.

## 5.2 Real Data Experiments

We used the UCLAF dataset as our real data experiment.

### 5.2.1 UCLAF Dataset

The UCLAF dataset (Zheng et al., 2010) is a commonly used benchmark dataset for coupled tensor completion (Ermis et al., 2015; Wimalawarne et al., 2018). The UCLAF dataset contains GPS data collected from $164$ users in $168$ locations performing $5$ activities. These GPS data forms a user-location-activity tensor $\mathcal{T} \in \mathbb{R}^{164 \times 168 \times 5}$ consisting of only a few observed elements. In order to learn the unobserved elements, tensor completion can be performed. However, the UCLAF dataset also contains side information that can be coupled to the tensor to improve the completion procedure. Similar to (Wimalawarne et al., 2018), we used the coupling of $\mathcal{T}$ with the user-location matrix $X \in \mathbb{R}^{164 \times 168}$. We used the same random data selection and validation processes as simulation experiments.

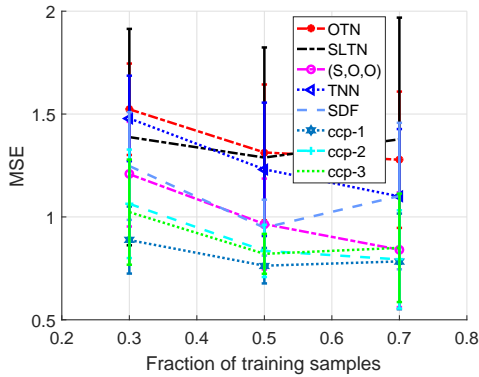

Figure 3: Performances on the UCLAF data set

Apart from the coupled nuclear norms, we experimented with the same baseline methods for tensors as in the previous section. For these experiments, we selected regularization parameters from logarithmic linear scale from $0.01$ to $5000$ with $200$ divisions. Additionally, we compared our results with the SDF model (Sorber et al., 2015) by using a CP rank of $2$.

Figure 3 shows that the coupled nuclear norm $\| \cdot \|_{\mathrm{ccp},(\lambda,\mathrm{F}),(\lambda,\mathrm{F})}$ (ccp-1) gives the best performance. The coupled norm $(\mathrm{S},\mathrm{O},\mathrm{O})$ which has given the best performance among multilinear rank based coupled norms (Wimalawarne et al., 2018) is outperformed by all the coupled nuclear norms.

Both simulation and UCLAF data experiments indicate that coupled nuclear norms lead to better performance compared to existing coupled norms (Wimalawarne et al., 2018).

## 6 Acknowledgment

This work has been partially supported by MEXT KAKENHI Grant Number 16H02868, Grant Number JPMJAC1503 ACCEL JST, FiDiPro Tekes (currently Business Finland) and AIPSE Academy of Finland.

## 7 Conclusion and Future Work

We introduce coupled nuclear norms by integrating the CP rank into coupled norms. We propose new coupled completion models regularized by coupled nuclear norms and discuss optimization procedures to solve them. Our excess risk bounds for coupled completion show that the proposed norms lead to better performances compared to existing multilinear rank based coupled norms. Our theoretical analysis is validated through simulation and real world data experiments, where we show that coupled nuclear norms can give better performance compared to existing methods. We believe that the proposed coupled nuclear norms should be further investigated to be widely applicable in real world problems.

Applying coupled nuclear norms to solve large scale problems is an important future research direction. More specifically, developing computationally feasible optimization methods is important since computing the coupled nuclear norms can be computationally costly. Future research in this direction can consider developing globally optimal power methods (Anandkumar et al., 2017) to approximate coupled nuclear norms. Furthermore, theoretical analysis of coupled nuclear norms with more than two tensors is another important future research direction.

## Footnotes

[1]Code and data are available at http://kishan-wimalawarne.com/onewebmedia/NeurIPS_2018_code.rar

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
