[Supplementary Material · supplementary.pdf]

# Appendix

## A Dual Norms of Coupled Nuclear Norms

In this section, we describe dual norms of coupled nuclear norms. Dual norms help us in developing optimization procedures and theoretical analysis.

The dual norm of the tensor nuclear norm for $\mathcal{T} \in \mathbb{R}^{n_1 \times \cdots \times n_K}$ (Yang et al., 2015; Yuan and Zhang, 2016) is defined as

$$\|\mathcal{T}\|_2 = \max_{\|y_i\|_2=1, 1 \leq i \leq K} \langle \mathcal{T}, y_1 \otimes y_2 \otimes \cdots \otimes y_K \rangle. \tag{9}$$

Below, we provide dual norms for the two coupled nuclear norms (2) and (3) with a $K$-mode tensor $\mathcal{W} \in \mathbb{R}^{n_1 \times \cdots \times n_K}$ and a $K'$-mode tensor $\mathcal{V} \in \mathbb{R}^{n'_1 \times \cdots \times n'_{K'}}$. These dual norms can be easily derived by taking spectral norms with respect to each tensor while considering the common factors on coupled modes.

**Theorem 3.** *The dual norm of* $\|\mathcal{W}, \mathcal{V}\|_{\mathrm{ccp},(\lambda_1,\mathrm{F})(\lambda_2,\mathrm{F})}$ *is*

$$\|\mathcal{W}, \mathcal{V}\|^a_{\mathrm{ccp},(\lambda_1,\mathrm{F})(\lambda_2,\mathrm{F})^\star} = \left\{ \lambda_1 \max_{\|x_l\|_2=1, l=1,\ldots,K} \langle \mathcal{W}, x_1 \otimes \cdots \otimes x_a \otimes \cdots x_K \rangle, \right.$$

$$\left. \lambda_2 \max_{\|y_{l'}\|_2=1, l'=1,\ldots,K' \backslash a} \langle \mathcal{V}, y_1 \otimes \cdots \otimes x_a \otimes \cdots y_{K'} \rangle \right\}. \tag{10}$$

*Proof.* By the (3), we have

$$\mathcal{W} = \sum_{i=1}^R \gamma_i x_{1i} \otimes \cdots \otimes x_{ai} \otimes \cdots x_{Ki},$$

and

$$\mathcal{V} = \sum_{i=1}^R \beta_i y_{1i} \otimes \cdots \otimes x_{ai} \otimes \cdots y_{K'i},$$

with $x_{ai}$, $i = 1, \ldots, R$ in common.

Taking the dual norms (spectral norms) (9) of them lead to

$$\|\mathcal{W}\|_2 = \max_{\|p_i\|_2=1, 1 \leq i \leq K} \langle \mathcal{W}, p_1 \otimes p_2 \otimes \cdots \otimes p_K \rangle,$$

and

$$\|\mathcal{V}\|_2 = \max_{\|q_i\|_2=1, 1 \leq i \leq K'} \langle \mathcal{V}, q_1 \otimes q_2 \otimes \cdots \otimes q_{K'} \rangle,$$

However, since by definition, each $x_{ai}$ is common to both tensors, $p_a = q_a$.

Further, due to $\|\mathcal{W}\|_* \leq \lambda_1$ and $\|\mathcal{V}\|_* \leq \lambda_2$ spectral norms of the coupled tensors also need to be scaled accordingly. This completes the proof. $\square$

We give following dual norm of (3) without proofs.

$$\inf_{\mathcal{W}^{(1)}+\mathcal{W}^{(2)}=\mathcal{W}} \|\mathcal{W}, \mathcal{V}\|^a_{\mathrm{ccp},(\lambda_1,\lambda_2,\mathrm{L}),(\lambda_3,\mathrm{F})^\star}$$

$$= \left\{ \lambda_1 \max_{\|x_l^{(1)}\|_2=1, l=1,\ldots,K; \|x_a\|_2=1} \langle \mathcal{W}^{(1)}, x_1^{(1)} \otimes \cdots \otimes x_a \otimes \cdots x_K^{(1)} \rangle, \right.$$

$$\lambda_2 \max_{\|x_l^{(2)}\|_2=1, l=1,\ldots,K} \langle \mathcal{W}^{(2)}, x_1^{(2)} \otimes \cdots \otimes x_K^{(2)} \rangle,$$

$$\left. \lambda_3 \max_{\|y_{l'}\|_2=1, l'=1,\ldots,K' \backslash a} \langle \mathcal{V}, y_1 \otimes \cdots \otimes x_a \otimes \cdots y_{K'} \rangle \right\}. \tag{11}$$

Dual norms for other coupled nuclear norms can be developed in a similar manner.

# B Optimization

In this section, we describe our extensions to the Frank-Wolfe optimization method to solve the proposed completion models specified in Section 3.1.

First, we point out that any coupled completion models (4) and (5) for partially observed tensors $\mathcal{X}$ and $\mathcal{Y}$ can be expressed as

$$\min_{\mathcal{W},\mathcal{V}\in\mathsf{W}} f_{\mathcal{W}}(\mathcal{X}) + f_{\mathcal{V}}(\mathcal{Y}), \tag{12}$$

where $\mathsf{W}$ is hypothesis class based on the coupled nuclear norm, and $f_{\mathcal{W}}(\cdot)$ and $f_{\mathcal{V}}(\cdot)$ are loss functions for $\mathcal{X}$ and $\mathcal{Y}$, respectively. We can convert (4) into (12) by taking $f_{\mathcal{W}}(\mathcal{X}) = \frac{1}{2}\|\Omega_1(\mathcal{W}) - \Omega_1(\mathcal{X})\|_{\mathrm{F}}^2$, $f_{\mathcal{V}}(\mathcal{Y}) = \frac{1}{2}\|\Omega_2(\mathcal{V}) - \Omega_2(\mathcal{Y})\|_{\mathrm{F}}^2$, and $\mathsf{W} = \{\mathcal{W}, \mathcal{V} \mid \|\mathcal{W}, \mathcal{V}\|_{\mathrm{ccp},(\lambda_1,\mathrm{F})(\lambda_2,\mathrm{F})}\}$. Similarly, we can convert (5) and other completion models regularized using coupled nuclear norms into the format of (12). An optimization problem with the formulation of (12) is solvable using the Frank-Wolfe method (Jaggi, 2013; Yang et al., 2015).

## B.1 Approximating the Coupled Spectral Norm

With the Frank-Wolfe optimization (Jaggi, 2013; Yang et al., 2015), learning models regularized by the nuclear norms only require to compute the spectral norm of tensors. To compute the spectral norm of a tensor, Yang et al. (2015) proposed an approximation method that recursively computes the largest singular vector of each mode. Their approximation method is listed in Algorithm 1. We use the build-in MATLAB functions: $[u, s, v] = \mathrm{svd}(U, 1)$ to find the largest singular vectors $u$ and $v$ and singular value $s$ of a matrix $U$, and $\mathrm{reshape}(u, [n_1, \dots, n_l])$ to reshape a vector $u \in \mathbb{R}^{n_1 n_2 \cdots n_l}$ to a tensor of dimensions $n_1 \times n_2 \times \cdots \times n_l$.

Lines 4-9 in ApproxSpectral() compute the topmost orthogonal vectors of a 4-mode tensor. This is based on the subroutine 1 of (Yang et al., 2015). In lines 11-16, it considers a $2^K$-mode tensor and recursively call itself to compute the topmost orthogonal vectors for the modes $1, \cdots, 2^{K-1}$ and using them computes the topmost orthogonal vectors of remaining modes. This is based on the subroutine 2 of (Yang et al., 2015).

1: **Input:** $\mathcal{A} \in \mathbb{R}^{n_1 \times n_2 \cdots \times n_K}$
2: **Output:** $x_1, \dots, x_K$
3: **if** $K == 4$ **then**
4:    $M = \mathrm{reshape}(\mathcal{A}, [n_1 n_2, n_3 n_4])$
5:    $(u, s, v) = \mathrm{svd}(M, 1)$
6:    $M_{left} = \mathrm{reshape}(u, [n_1, n_2])$
7:    $(x_1, s, x_2) = \mathrm{svd}(M_{left}, 1)$
8:    $M_{right} = M \times_1 x_1^\top \times_2 x_2^\top$
9:    $(x_3, s, x_4) = \mathrm{svd}(M_{right}, 1)$
10: **else**
11:    $\mathcal{A}' = \mathrm{reshape}(\mathcal{A}, [n_1 \cdots n_{2^{K-1}}, n_{2^{K-1}+1} \cdots n_{2^K}])$
12:    $u, s, v = \mathrm{svd}(\mathcal{A}', 1)$
13:    $\mathcal{A}_{left} = \mathrm{reshape}(u, [n_1, \dots, n_{2^{K-1}}])$
14:    $(x_1, \cdots, x_{2^{K-1}}) = \mathrm{ApproxSpectral}(\mathcal{A}_{left})$
15:    $\mathcal{A}_{right} = \mathcal{A} \times_1 x_1^\top \times_2 \cdots \times_{2^{K-1}} x_{2^{K-1}}^\top$
16:    $(x_{2^{K-1}+1}, \cdots, x_{2^K}) = \mathrm{ApproxSpectral}(\mathcal{A}_{right})$
17: **end if**

**Algorithm 1:** ApproxSpectral($\mathcal{A}$) based on (Yang et al., 2015)

We use the algorithm ArppoxSpectral(.) to approximate spectral norms of coupled nuclear norms (derived in the previous section) as given in Algorithm 2. In this algorithm, for simplicity, we consider only two tensors $\mathcal{A}$ and $\mathcal{B}$ that are coupled on the first mode. The line 4 in the Algorithm 2 computes the top left singular vector, $u_c$, of the concatenated matrix of unfolded $\mathcal{A}$ and $\mathcal{B}$ on the mode 1 which is common to both tensors. Lines 5 and 6, remove the first mode from the $\mathcal{A}$ and $\mathcal{B}$ using the common singular vector $u_c$, and in lines 7 and 8, singular vectors with respect to other modes are computed. Alternatively, we can use the Lanczos method instead of $\mathrm{svd}(\cdot, \cdot)$ in Algorithms 1 and 2.

1: **Input:** Tensor $\mathcal{A} \in \mathbb{R}^{n_1 \times n_2 \times \cdots \times n_K}$ and $\mathcal{B} \in \mathbb{R}^{n'_1 \times n'_2 \times \cdots \times n'_{K'}}$ coupled on mode-1
2: **Output:** $u_c, w_2, \ldots, w_K, v_2, \ldots, v_{K'}$
3: $Y = [A_{(1)}; B_{(1)}]$
4: $u_c, s, v = \mathrm{svd}(Y, 1)$
5: $\bar{\mathcal{A}} = \mathcal{A} \times_1 u_c$
6: $\bar{\mathcal{B}} = \mathcal{B} \times_1 u_c$
7: $w_2, \ldots, w_K = \mathrm{ApproxSpectral}(\bar{\mathcal{A}})$
8: $v_2, \ldots, v_{K'} = \mathrm{ApproxSpectral}(\bar{\mathcal{B}})$

**Algorithm 2:** ApproxCoupledSpectral($\mathcal{A}$, $\mathcal{B}$)

The proposed approximation method is convex, leading to a global solution for the proposed completion models (4) and (5). The approximation method can be further extended for multiple couplings of tensors by first computing singular vectors along all the coupled modes and then finding the largest singular vectors with respect to uncoupled modes. A limitation with the approximation method is the high computational cost in computing SVD for high dimensional coupled tensors.

## B.2 Optimization Procedure

In order to solve the proposed completion models, we extend the Frank-Wolfe optimization method proposed in (Yang et al., 2015). In Algorithm 3, we give procedures for solving the coupled model (4) regularized by $\|\mathcal{W}, \mathcal{V}\|_{\mathrm{ccp},(\lambda_1,\mathrm{F})(\lambda_2,\mathrm{F})}$. The most important step we want to highlight is the line 8, where the coupled factorization of the dual formulation of $\nabla f_{\mathcal{W}}(\mathcal{X})$ and $\nabla f_{\mathcal{V}}(\mathcal{Y})$ are obtained by using the Algorithm 1. Here, we use the coupled spectral norm (10) to update the gradient steps after factorizing the largest singular vectors including the common top most singular vector on mode 1. In lines 9 and 10, we compute projections of spectral norms of each tensor and update the learning models in lines 14 and 15.

1: **Input:** $\mathcal{X} \in \mathbb{R}^{n_1 \times n_2 \times \cdots \times n_K}$ with the mapping to the observed element by $\Omega_1$,
   $\mathcal{Y} \in \mathbb{R}^{n'_1 \times n'_2 \times \cdots \times n'_{K'}}$ with the mapping to the observed element by $\Omega_2$. Regularization parameters $\lambda_1$ and $\lambda_2$. Initial $\mathcal{W}^0$ and $\mathcal{V}^0$. Maximum number of iterations $T$.
2: **Output:** $\mathcal{W}^T$, $\mathcal{V}^T$
3: $t = 0$
4: **repeat**
5:    $t = t + 1$
6:    $f_{\mathcal{W}}(\mathcal{X}^t) = \frac{1}{2}\|\Omega_1(\mathcal{W}^t) - \Omega_1(\mathcal{X})\|_{\mathrm{F}}^2$
7:    $f_{\mathcal{V}}(\mathcal{Y}^t) = \frac{1}{2}\|\Omega_2(\mathcal{V}^t) - \Omega_2(\mathcal{Y})\|_{\mathrm{F}}^2$
8:    $u_c, w_2, \ldots, w_K, v_2, \ldots, v_{K'} = \mathrm{ApproxCoupledSpectral}(\nabla_{\mathcal{W}} f_{\mathcal{W}}(\mathcal{X}^t), \nabla_{\mathcal{V}} f_{\mathcal{V}}(\mathcal{Y}^t))$
9:    $\mathcal{W}^t_{descent} = -\lambda_1 u_c \otimes w_2, \cdots \otimes w_K$
10:    $\mathcal{V}^t_{descent} = -\lambda_2 u_c \otimes v_2, \cdots \otimes v_{K'}$
11:    **if** linesearch == True **then**
12:       Using an appropriate line search method (e.g.Yang et al., (2015))
13:    **else**
14:       $\mathcal{W}^{t+1} = \mathcal{W}^t + \frac{2}{t+2}\mathcal{W}^t_{descent}$
15:       $\mathcal{V}^{t+1} = \mathcal{V}^t + \frac{2}{t+2}\mathcal{V}^t_{descent}$
16:    **end if**
17: **until** t = T

**Algorithm 3:** A Frank-Wolfe optimization method for coupled tensors

We can extend the Algorithm 3 to optimize completion models that are regularized using other coupled nuclear norms.

# C   Proofs of Theoretical Analysis

In this section, we present the proofs of the theoretical results given in Section 4. To prove excess risk bounds, we need to know the expectation of the sum of spectral norms for coupled tensors, which we derive in the next subsection.

## C.1   Expectation of Coupled Spectral Norms

We recall that the spectral norm of a $K$-mode tensor $\mathcal{X} \in \mathbb{R}^{n \times \cdots \times n}$ is

$$\|\mathcal{X}\|_2 = \max_{\|y_i\|_2=1, 1\leq i\leq K} \langle \mathcal{X}, y_1 \otimes y_2 \otimes \cdots \otimes y_K \rangle,$$

and from (Nguyen et al., 2015) we know that it is equivalent to

$$\|\mathcal{X}\|_2 = \sup_{y_1, y_2, \ldots, y_{K-1} \in \mathbb{S}^n} \|\mathcal{X} \times_1 y_1 \times_2 \ldots \times_{K-1} y_{K-1}\|_2. \tag{13}$$

We use the definition of spectral norm in (13) to bound the summation of spectral norms of two coupled tensors. Though we can use tensor of any dimensions for our proof, it is often difficult to write with indexes for high dimensional tensors. For convenience, throughout our proofs, we use two tensors 3-mode tensors, $\mathcal{T} \in \mathbb{R}^{n \times n \times n}$ and $\mathcal{U} \in \mathbb{R}^{n \times n \times n}$, and describe them as $K$-mode and $K'$-modes tensors, respectively.

The next theorem gives the expectation of two coupled tensors.

**Theorem 4.** *Let $K$-mode tensor $\mathcal{T} \in \mathbb{R}^{n \times n \times \cdots \times n}$ and $K'$-mode tensor $\mathcal{U} \in \mathbb{R}^{n \times n \times \cdots \times n}$ are coupled on their first modes. We assume that entries of $\mathcal{T}$ and $\mathcal{U}$ are independent and zero mean. Given two positive values a and b, we have*

$$\mathbb{E}a\|\mathcal{T} \times_1 x \times_2 y\|_2 + b\|\mathcal{U} \times_1 x \times_2 z\|_2$$

$$\leq c_1\sqrt{2\pi}\left[a2^{3K+K'}\mathbb{E}_{\mathcal{T}}\alpha_1\left(\log_2 1/\eta\right)^{K-1} + b2^{K+3K'}\mathbb{E}_{\mathcal{U}}\alpha_2\left(\log_2 1/\eta\right)^{K'-1}\right.$$

$$\left. + a2^{K-1}\mathbb{E}_{\mathcal{T}}\beta_1\sqrt{\eta n} + b2^{K'-1}\mathbb{E}_{\mathcal{U}}\beta_2\sqrt{\eta n}\right]\sqrt{8(K+K'-3)\ln(5e/\eta)}, \tag{14}$$

*where*

$$\alpha_1^2 = \max_j\left(\max_{i_1,\ldots,i_{j-1},i_{j+1},\ldots,i_K}\left(\sum_{i_j}^n \mathcal{T}_{i_1,\ldots,i_{j-1},i_{j+1},\ldots,i_K}^2\right)\right),$$

$$\alpha_2^2 = \max_{j'}\left(\max_{i_1,\ldots,i_{j'-1},i_{j'+1},\ldots,i_{K'}}\left(\sum_{i_{j'}}^n \mathcal{U}_{i_1,\ldots,i_{j'-1},i_{j'+1},\ldots,i_{K'}}^2\right)\right),$$

$$\beta_1 = \max_{i_1,\ldots,i_K} |\mathcal{T}_{i_1,\ldots,i_K}|,$$

$$\beta_2 = \max_{i_1,\ldots,i_{K'}} |\mathcal{U}_{i_1,\ldots,i_{K'}}|,$$

*and $\eta$ is selected such that $\sqrt{(K+K'-3)\eta n \ln(5e/\eta)} \geq 1$.*

More importantly, we prove the next theorem which consider random tensors with elements from $\{0, -1, 1\}$, which is essential to prove excess risk bounds in Section 4.

**Theorem 5.** *Let $K$-mode tensor $\mathcal{T} \in \mathbb{R}^{n \times n \times \cdots \times n}$ and $K'$-mode tensor $\mathcal{U} \in \mathbb{R}^{n \times n \times \cdots \times n}$ are coupled on their first modes. We assume that entries of $\mathcal{T}$ and $\mathcal{U}$ are randomly sampled from the set $\{0, -1, 1\}$. By taking $\eta = (\ln n)^{2(\max(K,K')-1)}/n$, we have*

$$\mathbb{E}a\|\mathcal{T} \times_1 x \times_2 y\|_2 + b\|\mathcal{U} \times_1 x \times_2 z\|_2$$

$$\leq c_3\left[a2^{3K+K'}K\sqrt{n}(\ln n)^{K-1/2} + b2^{3K+K'}K'\sqrt{n}(\ln n)^{K'-1/2}\right],$$

*where a, b and $c_3$ are constants.*

To prove the above theorems, we use a similar approach as in (Nguyen et al., 2015), and use the *entropy-concentration tradeoff* analysis technique (Vershynin, 2011). The resulting proof is long, and to improve the readability we present our proof in several steps in the following subsections.

### C.1.1 Bounding by Gaussian Symmetrization

We use the well known Gaussian symmetrization results (Vershynin, 2011). Since $\mathcal{T}$ is a tensor with independent and zero mean entries, we have

$$\mathbb{E}_{\mathcal{T}}\|\mathcal{T}\|_2 = \mathbb{E}_{\mathcal{T}}\|\mathcal{T} - \mathcal{T}'\|_2,$$

where $\mathcal{T}'$ consists of independent random variables. Now from Jensen's inequality and introducing $\epsilon_{i,j,k} \in \{-1, 1\}$, we have

$$\mathbb{E}_{\mathcal{T}}\|\mathcal{T} - \mathcal{T}'\|_2 \leq \mathbb{E}_{\mathcal{T}}\mathbb{E}_{\epsilon}\left\|\sum_{i,j,k} \epsilon_{i,j,k}(\mathcal{T}_{i,j,k} - \mathcal{T}'_{ij,k})e_i \otimes e_j \otimes e_k\right\|_2$$

$$\leq 2\mathbb{E}_{\mathcal{T}}\mathbb{E}_{\epsilon}\left\|\sum_{i,j,k} \epsilon_{i,j,k}\mathcal{T}_{ij,k}e_i \otimes e_j \otimes e_k\right\|_2$$

Given that $g_{i,j,k}$ is sampled from a Gaussian distribution, we have

$$\mathbb{E}_{\mathcal{T}}\mathbb{E}_{g}\left\|g_{i,j,k}\mathcal{T}_{i,j,k}e_i \otimes e_j \otimes e_k\right\|_2 = \mathbb{E}_{\mathcal{T}}\mathbb{E}_{g}\mathbb{E}_{\epsilon}\left\|\sum_{i,l,k} \epsilon_{i,j,k}|g_{i,j,k}|\mathcal{T}_{i,j,k}e_i \otimes e_j \otimes e_k\right\|_2$$

$$\geq \mathbb{E}_{\mathcal{T}}\mathbb{E}_{\epsilon}\left\|\sum_{i,j,k} \mathbb{E}_{g}|g_{i,j,k}|\epsilon_{i,j,k}\mathcal{T}_{i,j,k}e_i \otimes e_j \otimes e_k\right\|_2$$

$$\geq \left(\frac{2}{\pi}\right)^{1/2}\mathbb{E}_{\mathcal{T}}\mathbb{E}_{\epsilon}\left\|\sum_{i,j,k} \epsilon_{i,j,k}\mathcal{T}_{i,j,k}e_i \otimes e_j \otimes e_k\right\|_2,$$

where we have used the fact that $\mathbb{E}|g_{i,j,k}| = \sqrt{2/\pi}$, and thus

$$\mathbb{E}_{\mathcal{T}}\|\mathcal{T}\|_2 \leq \sqrt{2\pi}\mathbb{E}_{\mathcal{T}}\mathbb{E}_{g}\left\|\sum_{i,j,k} g_{i,j,k}\mathcal{T}_{i,j,k}e_i \otimes e_j \otimes e_k\right\|_2. \tag{15}$$

Similarly, for the we have

$$\mathbb{E}_{\mathcal{U}}\|\mathcal{U}\|_2 \leq \sqrt{2\pi}\mathbb{E}_{\mathcal{U}}\mathbb{E}_{g'}\left\|\sum_{i,j,k} g'_{i,j,k}\mathcal{U}_{i,j,k}e_i \otimes e_j \otimes e_k\right\|_2, \tag{16}$$

where $g'_{i,j,k}$ is sampled from a Gaussian distribution.

In order to prove Theorem 1, we use following random tensors

$$\mathcal{H} = \sum_{i,j,k} g_{i,j,k}\mathcal{T}_{i,j,k}e_i \otimes e_j \otimes e_k, \tag{17}$$

and

$$\mathcal{G} = \sum_{i,j',k'} g'_{i,j',k'}\mathcal{U}_{i,j',k'}e_i \otimes e_{j'} \otimes e_{k'}. \tag{18}$$

Again we want to mention that we represent $\mathcal{H}$ and $\mathcal{G}$ as $K$-mode and $K'$-mode tensors, respectively, though they are 3-mode tensors.

### C.1.2 Bounding by Concentration Inequality

We now develop a concentration inequality to analyze the sum spectral norms of two coupled tensors.

Given a Lipschitz function $f : \mathbb{R}^n \mapsto \mathbb{R}$, the Lipschitz norm is defined (Nguyen et al., 2015) as

$$\|f\|_L = \sup_{x,y \in \mathbb{R}^n} \frac{|f(x) - f(y)|}{\|x - y\|_2}.$$

Additionally, we give the following well known results (Ledoux, 2001).

**Lemma 1.** *Let $f : \mathbb{R}^n \mapsto \mathbb{R}$ be a Lipschitz function and $\|f\|_L$ be its Lipschitz norm. Given a vector $g \in \mathbb{R}^n$ whose entries are a independent standard Gaussian random variables, then for all $t > 0$*

$$\mathbb{P}(f(g) \geq \mathbb{E}f(g) + t\sqrt{2}\|f\|_L) \leq e^{-t^2}. \tag{19}$$

We use the following result from (Nguyen et al., 2015) for (17) and (18) throughout our proof.

**Lemma 2.** *Given a pair of unit vectors $x$ and $y$*

$$\mathbb{E}_g \|\mathcal{H} \times_1 x \times_2 y\|_2 \leq \sqrt{\max_{i,j} \mathcal{A}_{i,j,k}^2}. \tag{20}$$

Using the above results we obtain the following lemma.

**Lemma 3.** *Given a unit vectors $x$, $y$, and $z$ and two positive values $a$ and $b$, we have*

$$\mathbb{P}\Bigg( a\|\mathcal{H} \times_1 x \times_2 y\|_2 + b\|\mathcal{G} \times_1 x \times_2 z\|_2 \geq a\sqrt{\max_{i,j}\sum_k \mathcal{T}_{i,j,k}} + b\sqrt{\max_{i,j'}\sum_{k'} \mathcal{U}_{i,j',k'}}$$

$$+ t2^{3/2} \max\left( a\max_k \left(\mathcal{T}_{i,j,k}^2 x_i^2 y_j^2\right)^{1/2} + b\max_{k'}\left(\mathcal{U}_{i,j',k'}^2 x_i^2 z_{j'}^2\right)^{1/2}\right)\Bigg) \leq e^{-t^2} \tag{21}$$

*Proof.* Given that $s_1 = \mathcal{H} \times_1 x \times_2 y$, where $\mathcal{H}$ is defined as (17), we know that

$$s_1 = \sum_{i,j,k}\left(\mathcal{H}_{ijk}x_iy_j\right)e_k$$

$$= \sum_k\left(\sum_{i,j}\mathcal{H}_{ijk}x_iy_j\right)e_k \tag{22}$$

$$= \sum_k\left(\sum_{i,j}g_{ijk}\mathcal{T}_{ijk}x_iy_j\right)e_k,$$

since $g_{ijk}$ is a Gaussian variable, $g_{ijk}\mathcal{T}_{ijk}x_iy_j$ also a random variable with zero mean and variance of $\sum_{i,j}\mathcal{T}_{ijk}^2 x_i^2 y_j^2$. Similarly, by considering $s_2 = \mathcal{G} \times_1 x \times_2 z$ with $\mathcal{G}$ as defined in (17), we obtain

$$s_2 = \sum_{k'}\left(\sum_{i,j'}g'_{ij'k'}\mathcal{U}_{ij'k'}x_iz_{j'}\right)e_{k'}, \tag{23}$$

with random variables $g'_{ij'k'}\mathcal{U}_{ij'k'}x_iz_{j'}$ having zero mean and a variance of $\mathcal{U}_{ij'k'}^2 x_i^2 z_{j'}^2$.

Let us consider

$$p_k^2 = \sum_{i,j}\mathcal{T}_{i,j,k}^2 x_i^2 y_j^2 \quad \text{for all } k \in [n],$$

and

$$q_{k'}^2 = \sum_{i,j'}\mathcal{U}_{i,j',k'}^2 x_i^2 z_{j'}^2 \quad \text{for all } k' \in [n].$$

Given $u \in \mathbb{R}^n$ and $v \in \mathbb{R}^n$, whose elements are standard Gaussian variables, we rewrite $s_1$ and $s_2$ as

$$s_1 = a\sum_k u_k p_k e_k,$$

and

$$s_2 = b\sum_{k'} v_{k'} q_{k'} e_{k'}.$$

Further, let us consider a concatenation of $s_1$ and $s_2$ as

$$s = [s_1^\top ; s_2^\top] = [a(u_1p_1, \ldots, u_np_n); b(v_1q_1, \ldots, v_nq_n)].$$

Now let us consider function $f$ as

$$f([u, v]) := a \left\| \sum_k u_k p_k e_k \right\|_2 + b \left\| \sum_{k'} v_{k'} q_{k'} e_{k'} \right\|_2,$$

then we have

$$f^2([u, v]) = a^2 \sum_k u_k^2 p_k^2 + b^2 \sum_{k'} v_{k'}^2 q_{k'}^2 + 2ab \left\| \sum_k u_k p_k e_k \right\|_2 \left\| \sum_{k'} v_{k'} q_{k'} e_{k'} \right\|_2.$$

We use the inequality $2xy \leq x^2 + y^2$ to obtain

$$f^2([u, v]) \leq 2 \left( a^2 \sum_k u_k^2 p_k^2 + b^2 \sum_{k'} v_{k'}^2 q_{k'}^2 \right) = 2\|s\|_2^2,$$

and

$$f^2([u, v]) \leq 2\|[u; v]\|_2^2 \max(a^2 \max_k p_k^2, b^2 \max_{k'} q_{k'}^2).$$

Thus

$$f([u, v]) \leq 2 \max \left( a \max_k p_k, b \max_{k'} q_{k'} \right) \leq 2 (a \max_k p_k + b \max_{k'} q_{k'}).$$

This leads to the Lipschitz norm of $f$ as

$$\|f\|_L = 2 \left( a \max_k \left( \mathcal{T}_{i,j,k}^2 x_i^2 y_j^2 \right)^{1/2} + b \max_{k'} \left( \mathcal{U}_{i,j',k'}^2 x_i^2 z_{j'}^2 \right)^{1/2} \right) \tag{24}$$

Finally, using the (24) and lemmas 1 and 2 we obtain the final bound. $\qquad \square$

Our goal is to bound the

$$\mathbb{E} \sup_{x,y,z \in \mathbb{S}^{n-1}} a \|\mathcal{H} \times_1 x \times_2 y\|_2 + b \|\mathcal{G} \times_1 x \times_2 z\|_2 \tag{25}$$

with the assistance of Lemma 3. Notice that $x$ is common both the spectral norms, which we know from Section A. Further, to use the entropy-concentration tradeoff method, we consider that each vector $x$, $y$, and $z$ have sparse components and spread components. Given $x = u + v$, where $u$ is the sparse component and $v$ is the spread component, and $\eta \in (0, 1]$, we have

$$u_i = \begin{cases} x_i & \text{if } |x_i| \geq \frac{1}{\sqrt{\eta n}}, \\ 0 & , \text{otherwise} \end{cases} \tag{26}$$

$$v_i = \begin{cases} x_i & \text{if } |x_i| < \frac{1}{\sqrt{\eta n}} \\ 0 & , \text{otherwise.} \end{cases} \tag{27}$$

Using the above we can obtain following two sets as in (Nguyen et al., 2015)

$$B_{2,0} = \left\{ x \in \mathbb{R}^n : \|x\|_2 \leq 1, |x_i| \geq \frac{1}{\sqrt{\eta n}} \text{ or } x_i = 0 \right\},$$

and

$$B_{2,\infty} = \left\{ x \in \mathbb{R}^n : \|x\|_2 \leq 1, \|x\|_\infty < \frac{1}{\sqrt{\eta n}} \right\}.$$

Using the fact $\mathbb{E}(p + q) = \mathbb{E}p + \mathbb{E}q$, we can expand (25) as

$$
\begin{aligned}
\mathbb{E} \sup_{x,y,z \in \mathbb{S}^{n-1}} a\|\mathcal{H} \times_1 x \times_2 y\|_2 + b\|\mathcal{G} \times_1 x \times_2 z\|_2 \leq &\ \mathbb{E} \sup_{x,y,z \in B_{2,0}} a\|\mathcal{H} \times_1 x \times_2 y\|_2 \\
&\qquad + b\|\mathcal{G} \times_1 x \times_2 z\|_2 \\
&+ \mathbb{E} \sup_{x,y,z \in B_{2,\infty}} a\|\mathcal{H} \times_1 x \times_2 y\|_2 \\
&\qquad + b\|\mathcal{G} \times_1 x \times_2 z\|_2 \\
&+ \mathbb{E} \sup_{x \in B_{2,0}} \sup_{y,z \in B_{2,\infty}} a\|\mathcal{H} \times_1 x \times_2 y\|_2 \\
&\qquad + b\|\mathcal{G} \times_1 x \times_2 z\|_2. \\
&+ \mathbb{E} \sup_{y \in B_{2,0}} \sup_{x,z \in B_{2,\infty}} a\|\mathcal{H} \times_1 x \times_2 y\|_2 \\
&\qquad + b\|\mathcal{G} \times_1 x \times_2 z\|_2 \\
&\qquad\qquad \vdots \\
&+ \mathbb{E} \sup_{x,y \in B_{2,0}} \sup_{z \in B_{2,\infty}} a\|\mathcal{H} \times_1 x \times_2 y\|_2 \\
&\qquad + b\|\mathcal{G} \times_1 x \times_2 z\|_2
\end{aligned}
\tag{28}
$$

In the subsequent sections we bound each of the right hand terms. We use following theorems from (Nguyen et al., 2015) to assist our proofs.

**Lemma 4.** *Let $X$ be a random variable assuming non-negative values. For all $t \geq 0$ and non-negative $h_1$, $h_2$, and $h_3$ if $\mathbb{P}(X \geq h_1 + th_2) \leq e^{-t^2+h_3}$, then, for all $q \geq 1$,*

$$
\mathbb{E}X^q \leq 3\sqrt{q}(h_1 + h_2\sqrt{h_3} + h_2\sqrt{q/2})^q.
$$

**Lemma 5.** *Let $\mathbb{N}$ be an $\epsilon$-net for a $B$ associated with a norm $\|\cdot\|_2$. Then, the spectral norm of a $d$-mode tensor $\mathcal{A}$ is bounded by,*

$$
\sup_{x_1 \cdots x_{d-1} \in B} \|\mathcal{A} \times_1 x_1 \cdots \times_{d-1} x_{d-1}\|_2 \leq \left(\frac{1}{1-\epsilon}\right)^{d-1} \sup_{x_1 \cdots x_{d-1} \in \mathbb{N}} \|\mathcal{A} \times_1 x_1 \cdots \times_{d-1} x_{d-1}\|_2.
$$

### C.1.3 Control of Sparse Vectors

We prove following lemma in this section.

**Lemma 6.** *Let us consider a $K$-mode tensor $\mathcal{T} \in \mathbb{R}^{n \times \cdots \times n}$ with its Gaussian symmetrization $\mathcal{H}$ defined in (17) and a $K'$-mode tensor $\mathcal{U} \in \mathbb{R}^{n \times \cdots \times n}$ with its Gaussian symmetrization $\mathcal{G}$ defined in (18). Let us define the following*

$$
\alpha_1^2 = \max\left\{ \max_{i,j} \sum_{k=1}^n \mathcal{T}_{ijk}^2, \max_{i,k} \sum_{j=1}^n \mathcal{T}_{ijk}^2, \max_{j,k} \sum_{i=1}^n \mathcal{T}_{ijk}^2 \right\},
\tag{29}
$$

$$
\beta_1 = \max_{ijk} |\mathcal{T}_{ijk}|,
\tag{30}
$$

$$
\alpha_2^2 = \max\left\{ \max_{i,j'} \sum_{k'=1}^n \mathcal{U}_{ij'k'}^2, \max_{i,k'} \sum_{j'=1}^n \mathcal{U}_{ij'k'}^2, \max_{j',k'} \sum_{i=1}^n \mathcal{U}_{ij'k'}^2 \right\},
\tag{31}
$$

$$
\beta_2 = \max_{ij'k'} |\mathcal{U}_{ij'k'}|,
\tag{32}
$$

*then*

$$\mathbb{E}\left( \sup_{x,y,z\in B_{2,0}} a\|\mathcal{H}\times_1 x\times_2 y\|_2 + b\|\mathcal{G}\times_1 x\times_2 z\|_2 \geq 2^{K-1}\alpha_1 + 2^{K'-1}\alpha_2 \right.$$

$$+ 2^{3/2}t\left(a2^{K-1}\beta_1 + b2^{K'-1}\beta_2\right)\bigg)$$

$$\leq 3\left(2^{K-1}\left(\alpha_1 + 2^{3/2}\beta_1\left(\sqrt{(K+K'-3)\eta n\ln(5e/\eta)} + 1\right)\right)\right.$$

$$+ 2^{K'-1}\left(\alpha_2 + 2^{3/2}\beta_2\left(\sqrt{(K+K'-3)\eta n\ln(5e/\eta)} + 1\right)\right)\bigg)$$

*Proof.* Let $D = \eta n$ and $B_{2,0,D}$ be a $D$-dimensional set defined as

$$B_{2,0,D} = \{x \in \mathbb{R}^D : \|x\|_2 \leq 1\}. \tag{33}$$

The set $B_{2,0}$ represent the set of vectors with at most $D$ non-zero entries such that $B_{2,0} = \cup B_{2,0,D}$ (Nguyen et al., 2015). We also know from (Nguyen et al., 2015) that there are at most $\binom{n}{D} \leq \left(\frac{en}{D}\right)^D$ sets of $B_{2,0,D}$. Further, we know that the $1/2$-net of a subset $B_{2,0,D}$, $N_{B_{2,0,D}}$, has a cardinality that is bounded by $5^D$ (Nguyen et al., 2015).

Using Lemma 5 with $\epsilon = 1/2$, we obtain

$$\sup_{x,y\in B_{2,0,D}} \|\mathcal{H}\times_1 x\times_2 y\|_2 \leq 2^{K-1} \sup_{x,y\in N_{B_{2,0,D}}} \|\mathcal{H}\times_1 x\times_2 y\|_2,$$

and similarly

$$\sup_{x,z\in B_{2,0,D}} \|\mathcal{G}\times_1 x\times_2 z\|_2 \leq 2^{K'-1} \sup_{x,z\in N_{B_{2,0,D}}} \|\mathcal{G}\times_1 x\times_2 z\|_2.$$

We also can bound the following

$$\max\left(\sum_{i,j}\mathcal{T}_{i,j,k}^2 x_i^2 y_j^2\right)^{1/2} \leq \max_{i,j,k}|\mathcal{T}_{i,j,k}^2|\left(\sum_{i,j}x_i^2 y_j^2\right)^{1/2} \leq \max_{i,j,k}|\mathcal{T}_{i,j,k}^2| = \beta_1,$$

and

$$\max\left(\sum_{i,j'}\mathcal{U}_{i,j',k'}^2 x_i^2 z_{j'}^2\right)^{1/2} \leq \max_{i,j',k'}|\mathcal{U}_{i,j',k'}^2|\left(\sum_{i,j'}x_i^2 z_{j'}^2\right)^{1/2} \leq \max_{i,j',k'}|\mathcal{U}_{i,j',k'}^2| = \beta_2.$$

Applying the above bound to our concentration inequality in Lemma 3 and taking the union bound on all possible combination over vectors $x$, $y$, and $z$ we obtain

$$\mathbb{P}\left( \sup_{x,y,z\in B_{2,0,D}} a\|\mathcal{H}\times_1 x\times_2 y\|_2 + b\|\mathcal{G}\times_1 x\times_2 z\|_2 \geq a2^{K-1}\alpha_1 + b2^{K'-1}\alpha_2 \right.$$

$$+ 2^{3/2}t\left(a2^{K-1}\beta_1 + b2^{K'-1}\beta_2\right)\bigg)$$

$$\leq (5^D)^{(K-1)}(5^D)^{(K'-2)}e^{-t^2} = (5^D)^{(K+K'-3)}e^{-t^2},$$

where $\alpha_1$, $\alpha_2$, $\beta_1$, and $\beta_1$ are defined as in (29), (31), (30), and (32), respectively.

In the above bound, we emphasize that $x$ is common to both $\mathcal{H}$ and $\mathcal{G}$ and it leads to a bounding by $(5^D)^{(K+K'-3)}$ from the union bound. Taking union bound with respect to all possible $B_{2,0,D}$ sets of $B_{2,0}$, we have

$$\mathbb{P}\left( \sup_{x,y,z\in B_{2,0}} a\|\mathcal{H}\times_1 x\times_2 y\|_2 + b\|\mathcal{G}\times_1 x\times_2 z\|_2 \geq a2^{K-1}\alpha_1 + b2^{K'-1}\alpha_2 \right.$$

$$+ 2^{3/2}t\left(a2^{K-1}\beta_1 + b2^{K'-1}\beta_2\right)\bigg)$$

$$\leq \left(\left(\frac{en}{D}\right)^D\right)^{(K+K'-3)}(5^D)^{(K+K'-3)}e^{-t^2} = \left(\frac{5e}{\eta}\right)^{\eta n(K+K'-3)}e^{-t^2}.$$

Finally, we apply the Lemma 4 with $q = 1$, $h_1 = a2^{K-1}\alpha_1 + b2^{K'-1}\alpha_2$, $h_2 = 2^{3/2}(a2^{K-1}\beta_1 + b2^{K'-1}\beta_2)$, and $h_3 = (K + K' - 3)\eta n \ln(5e/\eta)$ to obtain the final bound

$$
\mathbb{E}\left( \sup_{x,y,z \in B_{2,0}} a\|\mathcal{H} \times_1 x \times_2 y\|_2 + b\|\mathcal{G} \times_1 x \times_2 z\|_2 \geq 2^{K-1}\alpha_1 + 2^{K'-1}\alpha_2 \right.
$$

$$
+ 2^{3/2}t\left( a2^{K-1}\beta_1 + b2^{K'-1}\beta_2 \right) \bigg)
$$

$$
\leq 3\Big( 2^{K-1}\Big( \alpha_1 + 2^{3/2}\beta_1\big( \sqrt{(K + K' - 3)\eta n \ln(5e/\eta)} + 1\big)\Big)
$$

$$
\left. + 2^{K'-1}\Big( \alpha_2 + 2^{3/2}\beta_2\big( \sqrt{(K + K' - 3)\eta n \ln(5e/\eta)} + 1\big)\Big)\right)
$$

$\square$

### C.1.4 Control of Spread Vectors

Now we bound the spread vectors with respect to the set $B_{2,\infty}$.

**Lemma 7.** *Let us consider a $K$-mode tensor $\mathcal{T} \in \mathbb{R}^{n \times \cdots \times n}$ with its Gaussian symmetrization $\mathcal{H}$ defined as (17) and a $K'$-mode tensor $\mathcal{U} \in \mathbb{R}^{n \times \cdots \times n}$ with its Gaussian symmetrization $\mathcal{G}$ defined as (18). Given (29), (30), (31), and (32) we have*

$$
\mathbb{E} \sup_{x,y,z \in B_{2,\infty}} a\|\mathcal{H} \times_1 x \times_2 y\|_2 + b\|\mathcal{G} \times_1 x \times_2 z\|_2
$$

$$
\leq 3\left( 4^{K-1}(\log_2 1/\eta)^{K-1}\left( a\alpha_1 + 2^{3/2}a\alpha_1\left( \sqrt{(K + K' - 3)\ln(5e/\eta)} + \frac{1}{\sqrt{\eta n}}\right)\right)\right.
$$

$$
\left. + 4^{K'-1}(\log_2 1/\eta)^{K'-1}\left( b\alpha_2 + 2^{3/2}b\alpha_2\left( \sqrt{(K + K' - 3)\ln(5e/\eta)} + \frac{1}{\sqrt{\eta n}}\right)\right)\right).
$$

*Proof.* To prove this theorem, we need to bound with respect to the $\epsilon$-net of $B_{2,\infty}$. Following (Nguyen et al., 2015), we have the set of vectors $N_k$ for $k = 0, 1, \ldots, 2M - 1$ where $M = \lceil 2 + \log_2 1/\sqrt{\eta}\rceil$ as

$$
N_k = \left\{ v \in B_{2,\infty} : \text{for all } i \in [n],\ v_i = \pm\frac{1}{2^{k/2}\sqrt{\eta n}} \text{ or } v_i = 0\right\},
$$

and the $1/2$-net of $B_{2,\infty}$ (Lemma 9 of (Nguyen et al., 2015)) as

$$
N_{B_{2,\infty}} = \left\{ v \in B_{2,\infty} : \forall i \in [n], v_i = \pm\frac{1}{2^{k/2}\sqrt{\eta n}} \text{ with either } k = 0, 1, \ldots, 2M - 1 \text{ or } v_i = 0\right\}.
$$

To obtain our result, we use the concentration inequality in Lemma 3. We take $\|x\|_2 \leq 1$ and $\|y\|_\infty = \frac{1}{2^k\sqrt{\eta n}}$, then we have

$$
\max_l\left( \sum_{i,j} \mathcal{T}_{ijl}^2 x_i^2 y_j^2\right) = \max_l\left( \sum_i x_i^2 \sum_j \mathcal{T}_{ijl}^2 y_j^2\right)
$$

$$
\leq \max_l \frac{1}{2^k\eta n}\left( \sum_j y_j^2 \sum_i \mathcal{T}_{ijl}^2\right)
$$

$$
\leq \max_l \frac{1}{2^k\eta n} \max_{i,j} \sum_i \mathcal{T}_{ijl}^2
$$

Similarly, we have

$$\max_{l'}\left(\sum_{i,j'}\mathcal{U}_{ij'l'}^2 x_i^2 z_{j'}^2\right) = \max_{l'}\left(\sum_i x_i^2 \sum_{j'}\mathcal{U}_{ij'l'}^2 z_{j'}^2\right)$$

$$\leq \max_{l'}\frac{1}{2^k\eta n}\left(\sum_{j'}z_{j'}^2\sum_i \mathcal{U}_{ij'l'}^2\right)$$

$$\leq \max_{l'}\frac{1}{2^k\eta n}\max_{i,j'}\sum_i \mathcal{U}_{ij'l'}^2$$

Now using the Lemma 3, we obtain

$$\mathbb{P}\left(a\|\mathcal{H}\times_1 x\times_2 y\|_2 + b\|\mathcal{G}\times_1 x\times_2 z\|_2 \geq a2^{K-1}\alpha_1 + b2^{K'-1}\alpha_2 \right.$$
$$\left. + t\frac{2^{3/2}}{2^{k/2}\sqrt{\eta n}}\left(a2^{K-1}\alpha_1 + b2^{K'-1}\alpha_2\right)\right) \leq e^{-t^2}.$$

From Lemma 12 of (Nguyen et al., 2015), we know that $|N_k| \leq e^{2^k\eta n\ln(2e/\eta)}$ and using this bounds we apply union bound on all possible combination of $N_k$, $N_{k'}$, and $N_{k''}$ which results in

$$\mathbb{P}\left(\sup_{x\in N_k, y\in N_{k'}, z\in N_{k''}} a\|\mathcal{H}\times_1 x\times_2 y\|_2 + b\|\mathcal{G}\times_1 x\times_2 z\|_2 \geq a2^{K-1}\alpha_1 + b2^{K'-1}\alpha_2 \right.$$
$$\left. + t\frac{2^{3/2}}{2^{k/2}\sqrt{\eta n}}\left(a2^{K-1}\alpha_1 + b2^{K'-1}\alpha_2\right)\right) \leq e^{-t^2+(K+K'-3)2^k\eta n\ln(5e/\eta)}$$

We apply the Lemma 4 with $q = 1$, $h_1 = a2^{K-1}\alpha_1 + b2^{K'-1}\alpha_2$, $h_2 = 2^{3/2}(a2^{K-1}\alpha_1 + b2^{K'-1}\alpha_2)$, and $h_3 = (K+K'-3)2^{k/2}\ln(5e/\eta)$ to obtain

$$\mathbb{E}\sup_{x\in N_k, y\in N_{k'}, z\in N_{k''}}\|\mathcal{H}\times_1 x\times_2 y\|_2 + \|\mathcal{G}\times_1 x\times_2 z\|_2$$

$$\leq 3\left(a2^{K-1}\alpha_1 + b2^{K'-1}\alpha_2 + 2^{3/2}(a2^{K-1}\alpha_1 + b2^{K'-1}\alpha_2)\sqrt{(K+K'-3)\ln(5e/\eta)}\right.$$

$$\left. + 2^{3/2}(a2^{K-1}\alpha_1 + b2^{K'-1}\alpha_2)\sqrt{\frac{1}{2^k\eta n}}\right)$$

Summing all the possibilities sets of $N_k$, $N_{k'}$, and $N_{k''}$ of $B_{2,\infty}$, we obtain

$$\mathbb{E}\sup_{x,y,z\in B_{2,\infty}} a\|\mathcal{H}\times_1 x\times_2 y\|_2 + b\|\mathcal{G}\times_1 x\times_2 z\|_2$$

$$\leq 3\left(\sum_{k=0}^{2M-1}\sum_{k'=0}^{2M-1}\left(a2^{K-1}\alpha_1 + a2^{K-1}\alpha_1\sqrt{2(K+K'-3)\ln(5e/\eta)}\right)\right.$$

$$+ \sum_{k=0}^{2M-1}\sum_{k''=0}^{2M-1}\left(b2^{K'-1}\alpha_2 + b2^{K'-1}\alpha_2\sqrt{2(K+K'-3)\ln(5e/\eta)}\right)$$

$$\left. + 2^{3/2}\left(\sum_{k=0}^{2M-1}\sum_{k'=0}^{2M-1}a2^{K-1}\alpha_1 + \sum_{k=0}^{2M-1}\sum_{k''=0}^{2M-1}b2^{K'-1}\alpha_2\right)\sqrt{\frac{1}{2^k\eta n}}\right),$$

which can be simplified to

$$
\mathbb{E} \sup_{x,y,z \in B_{2,\infty}} a\|\mathcal{H} \times_1 x \times_2 y\|_2 + b\|\mathcal{G} \times_1 x \times_2 z\|_2
$$

$$
\leq 3 \Bigg( (2M)^2 \Big( a2^{K-1}\alpha_1 + 2^{3/2} a2^{K-1}\alpha_1 \sqrt{2(K+K'-3)\ln(5e/\eta)} \Big)
$$

$$
+ (2M)^2 \Big( b2^{K'-1}\alpha_2 + 2^{3/2} b2^{K'-1}\alpha_2 \sqrt{(K+K'-3)\ln(5e/\eta)} \Big)
$$

$$
+ 2^{3/2} \Big( (2M)^2 a2^{K-1}\alpha_1 + (2M)^2 b2^{K'-1}\alpha_2 \Big) \sqrt{\frac{1}{2^k \eta n}} \Bigg),
$$

and taking summation over $K$-mode and $K'$-mode tensors we obtain

$$
\mathbb{E} \sup_{x,y,z \in B_{2,\infty}} a\|\mathcal{H} \times_1 x \times_2 y\|_2 + b\|\mathcal{G} \times_1 x \times_2 z\|_2
$$

$$
\leq 3 \Bigg( (2M)^{K-1} \Big( a2^{K-1}\alpha_1 + 2^{3/2} a2^{K-1}\alpha_1 \sqrt{(K+K'-3)\ln(5e/\eta)} \Big)
$$

$$
+ (2M)^{K'-1} \Big( b2^{K'-1}\alpha_2 + 2^{3/2} b2^{K'-1}\alpha_2 \sqrt{(K+K'-3)\ln(5e/\eta)} \Big)
$$

$$
+ 2^{3/2} \Big( (2M)^{K-1} a2^{K-1}\alpha_1 + (2M)^{K'-1} b2^{K'-1}\alpha_2 \Big) \sqrt{\frac{1}{2^k \eta n}} \Bigg).
$$

Since $M = \lceil 2 + \log_2 1/\sqrt{\eta} \rceil \leq \log_2 1/\eta$ (Nguyen et al., 2015) and $2^k \geq 1$ for $k \geq 0$, we obtain the final bound

$$
\mathbb{E} \sup_{x,y,z \in B_{2,\infty}} a\|\mathcal{H} \times_1 x \times_2 y\|_2 + b\|\mathcal{G} \times_1 x \times_2 z\|_2
$$

$$
\leq 3 \Bigg( 4^{K-1}(\log_2 1/\eta)^{K-1} \Big( a\alpha_1 + 2^{3/2} a\alpha_1 \Big( \sqrt{(K+K'-3)\ln(5e/\eta)} + \frac{1}{\sqrt{\eta n}} \Big) \Big)
$$

$$
+ 4^{K'-1}(\log_2 1/\eta)^{K'-1} \Big( b\alpha_2 + 2^{3/2} b\alpha_2 \Big( \sqrt{(K+K'-3)\ln(5e/\eta)} + \frac{1}{\sqrt{\eta n}} \Big) \Big) \Bigg).
$$

$\square$

### C.1.5 Control over both Sparse and Spread Vectors

We now consider case where we have vectors from both $B_{2,0}$ and $B_{2,\infty}$. We prove the following lemma.

**Lemma 8.** *Let us consider a $K$-mode tensor $\mathcal{T} \in \mathbb{R}^{n \times \cdots \times n}$ with its Gaussian symmetrization $\mathcal{H}$ defined as (17) and a $K'$-mode tensor $\mathcal{U} \in \mathbb{R}^{n \times \cdots \times n}$ with its Gaussian symmetrization $\mathcal{G}$ defined as (18). Given (29), (30) ,(31) ,and (32) we have*

$$
\mathbb{E} \sup_{x \in B_{2,0}, y,z \in B_{2,\infty}} a\|\mathcal{H} \times_1 x \times_2 y\|_2 + b\|\mathcal{G} \times_1 x \times_2 z\|_2
$$

$$
\leq 3 \Bigg( 4^{K-1}(\log_2 1/\eta)^{K-1} \Big( a\alpha_1 + 2^{3/2} a\alpha_1 \Big( \sqrt{(K+K'-3)\ln(5e/\eta)} + \frac{1}{\sqrt{\eta n}} \Big) \Big)
$$

$$
+ 4^{K'-1}(\log_2 1/\eta)^{K'-1} \Big( b\alpha_2 + 2^{3/2} b\alpha_2 \Big( \sqrt{(K+K'-3)\ln(5e/\eta)} + \frac{1}{\sqrt{\eta n}} \Big) \Big) \Bigg).
$$

*Proof.* To prove this lemma, we use same arguments as used in Lemma 14 of (Nguyen et al., 2015). Here, we want to bound the coupled spectral norm using a mixture of $B_{2,0}$ and $B_{2,\infty}$. In the previous

two lemmas, we defined $N_{B_{2,0}}$ and $N_{B_{2,\infty}}$ as $1/2$-nets for $B_{2,0}$ and $B_{2,\infty}$, respectively. Further, we use the fact that cardinality of upper bound of set $N_{B_{2,\infty}}$ is larger than cardinality of upper bound of set $N_{B_{2,0}}$ (Nguyen et al., 2015). This allows us to consider the worse case scenario, where one vector from $x$, $y$, and $z$ belongs to $N_{B_{2,0}}$ and the rest to $N_{B_{2,\infty}}$.

Without losing generality, we take $x \in N_{B_{2,0}}$ and $y, z \in N_{B_{2,\infty}}$. Then

$$\mathbb{P}\Bigg(a\|\mathcal{H} \times_1 x \times_2 y\|_2 + b\|\mathcal{G} \times_1 x \times_2 z\|_2 \geq a2^{K-1}\alpha_1 + b2^{K'-1}\alpha_2$$
$$+ t\frac{2^{3/2}}{2^{\max\{k,\ldots,k'\}/2}\sqrt{\eta n}}\big(a2^{K-1}\alpha_1 + b2^{K'-1}\alpha_2\big)\Bigg) \leq e^{-t^2},$$

where used $\|x\|_2 \leq 1$ and selected minimum among $\|y\|_\infty \leq 1/2^{k/2}$ and $\|z\|_\infty \leq 1/2^{k'/2}$.

Now, taking the union bound with respect to $x \in N_{B_{2,0}}$ and $y, z \in N_{B_{2,\infty}}$, and following a similar approach as in Lemma 14 of (Nguyen et al., 2015), we obtain

$$\mathbb{P}\Bigg(\sup_{x \in N_{B_{2,0}}, y \in N_k, z \in N_{k'}} a\|\mathcal{H} \times_1 x \times_2 y\|_2 + b\|\mathcal{G} \times_1 x \times_2 z\|_2 \geq a2^{K-1}\alpha_1 + b2^{K'-1}\alpha_2$$
$$+ t\frac{2^{3/2}}{2^{\max\{k,\ldots,k'\}/2}\sqrt{\eta n}}\big(a2^{K-1}\alpha_1 + b2^{K'-1}\alpha_2\big)\Bigg) \leq e^{-t^2 + (K+K'-3)2^{\max\{k,\ldots,k'\}}\eta n \ln(5e/\eta)}.$$

Applying Lemma 4 , with $q = 1$, $h_1 = a2^{K-1}\alpha_1 + b2^{K'-1}\alpha_2$, $h_2 = \frac{2^{3/2}}{2^{\max\{k,\ldots,k'\}/2}\sqrt{\eta n}}(a2^{K-1}\alpha_1 + b2^{K'-1}\alpha_2)$, and $h_3 = 2(K + K' - 3)2^{\max\{k,\ldots,k'\}}\ln(5e/\eta)$, leads to

$$\mathbb{E}\sup_{x \in N_{B_{2,0}}, y \in N_k, z \in N_{k'}} a\|\mathcal{H} \times_1 x \times_2 y\|_2 + b\|\mathcal{G} \times_1 x \times_2 z\|_2 \leq 3\Bigg(a2^{K-1}\alpha_1 + b2^{K'-1}\alpha_2$$
$$+ 2^{3/2}\Big(a2^{K-1}\alpha_1 + b2^{K'-1}\alpha_2\Big)\sqrt{2(K+K'-3)\ln(5e/\eta)}$$
$$+ \frac{2^{3/2}}{\sqrt{\eta n}}\Big(\frac{1}{2^{\max\{k,\cdots,k'\}/2}}a2^{K-1}\alpha_1 + \frac{1}{2^{\max\{k,\cdots,k'\}/2}}b2^{K'-1}\alpha_2\Big)\Bigg).$$

Bounding over the sets $x \in B_{2,0}$, $y \in B_{2,\infty}$, and $z \in B_{2,\infty}$, and $M \leq \log_2 1/\eta$ we obtain

$$\mathbb{E}\sup_{x \in B_{2,0}y, z \in B_{2,\infty}} a\|\mathcal{H} \times_1 x \times_2 y\|_2 + b\|\mathcal{G} \times_1 x \times_2 z\|_2$$
$$\leq 3\Bigg(4^{K-1}(\log_2 1/\eta)^{K-2}\Big(a\alpha_1 + 2^{3/2}a\alpha_1\Big(\sqrt{(K+K'-3)\ln(5e/\eta)} + \frac{1}{2^{\max\{k,\cdots,k'\}/2}\sqrt{\eta n}}\Big)\Big)$$
$$+ 4^{K'-1}(\log_2 1/\eta)^{K'-2}\Big(b\alpha_1 + 2^{3/2}b\alpha_1\Big(\sqrt{(K+K'-3)\ln(5e/\eta)} + \frac{1}{2^{\max\{k,\cdots,k'\}/2}\sqrt{\eta n}}\Big)\Big)\Bigg).$$

Since $2^{\max\{k,\cdots,k'\}} \geq 1$, we obtain

$$\mathbb{E}\sup_{x \in B_{2,0}, y, z \in B_{2,\infty}} a\|\mathcal{H} \times_1 x \times_2 y\|_2 + b\|\mathcal{G} \times_1 x \times_2 z\|_2$$
$$\leq 3\Bigg(4^{K-1}(\log_2 1/\eta)^{K-2}\Big(a\alpha_1 + 2^{3/2}a\alpha_1\Big(\sqrt{(K+K'-3)\ln(5e/\eta)} + \frac{1}{\sqrt{\eta n}}\Big)\Big)$$
$$+ 4^{K'-1}(\log_2 1/\eta)^{K'-2}\Big(b\alpha_2 + 2^{3/2}b\alpha_2\Big(\sqrt{(K+K'-3)\ln(5e/\eta)} + \frac{1}{\sqrt{\eta n}}\Big)\Big)\Bigg).$$

$\square$

### C.1.6 Final Bounds

We now prove the Theorem 4 using the above lemmas.

*Proof of Theorem 4.* We combine results from Lemmas 6,7, and 8 with (28) to obtain

$$\mathbb{E}a\|\mathcal{H}\times_1 x\times_2 y\|_2 + b\|\mathcal{G}\times_1 x\times_2 z\|_2$$

$$\leq 3\bigg(a2^{K-1}\Big(\alpha_1 + 2^{3/2}\beta_1\big(\sqrt{(K+K'-3)\eta n\ln(5e/\eta)} + 1\big)\Big)$$

$$+ b2^{K'-1}\Big(\alpha_2 + 2^{3/2}\beta_2\big(\sqrt{(K+K'-3)\eta n\ln(5e/\eta)} + 1\big)\Big)$$

$$+ a4^{K-1}(\log_2 1/\eta)^{K-1}\Big(\alpha_1 + 2^{3/2}\alpha_1\Big(\sqrt{(K+K'-3)\ln(5e/\eta)} + \frac{1}{\sqrt{\eta n}}\Big)\Big)$$

$$+ b4^{K'-1}(\log_2 1/\eta)^{K'-1}\Big(\alpha_2 + 2^{3/2}\alpha_2\Big(\sqrt{(K+K'-3)\ln(5e/\eta)} + \frac{1}{\sqrt{\eta n}}\Big)\Big)$$

$$+ a(2^{K+K'-2}-2)\times 4^{K-1}(\log_2 1/\eta)^{K-2}\Big(\alpha_1 + 2^{3/2}\alpha_1\Big(\sqrt{(K+K'-3)\ln(5e/\eta)} + \frac{1}{\sqrt{\eta n}}\Big)\Big)$$

$$+ b(2^{K+K'-2}-2)\times 4^{K'-1}(\log_2 1/\eta)^{K'-2}\Big(\alpha_2 + 2^{3/2}\alpha_2\Big(\sqrt{(K+K'-3)\ln(5e/\eta)} + \frac{1}{\sqrt{\eta n}}\Big)\Big),$$

where the last two summations are results from taking different combinations for the coupled $K$-mode and $K'$-mode tensors, which results in $2^{K+K'-2}-2$ combinations of the result in Lemma 8.

Given that $\sqrt{(K+K'-3)\eta n\ln(5e/\eta)} \geq 1$, we have

$$\mathbb{E}a\|\mathcal{T}\times_1 x\times_2 y\|_2 + b\|\mathcal{U}\times_1 x\times_2 z\|_2$$

$$\leq c_1\sqrt{2\pi}\bigg[a2^{3K+K'}\mathbb{E}_{\mathcal{T}}\alpha_1\Big(\log_2 1/\eta\Big)^{K-1} + b2^{K+3K'}\mathbb{E}_{\mathcal{U}}\alpha_2\Big(\log_2 1/\eta\Big)^{K'-1}$$

$$+ a2^{K-1}\mathbb{E}_{\mathcal{T}}\beta_1\sqrt{\eta n} + b2^{K'-1}\mathbb{E}_{\mathcal{U}}\beta_2\sqrt{\eta n}\bigg]\sqrt{8(K+K'-3)\ln(5e/\eta)}.$$

$$\square$$

Finally, we prove the Theorem 5.

*Proof of Theorem 5.* Since all elements of the tensors are from $\{-1,0,1\}$, the definitions (30) and (32) lead to

$$\mathbb{E}_{\mathcal{T}}\beta_1 = \mathbb{E}_{\mathcal{T}}\max_{ijk}|\mathcal{T}_{ijk}| = 1,$$

and

$$\mathbb{E}_{\mathcal{U}}\beta_2 = \mathbb{E}_{\mathcal{U}}\max_{ijk}|\mathcal{U}_{ijk}| = 1.$$

Further, we find that (29) and (31) are

$$\mathbb{E}_{\mathcal{T}}\alpha_1 = \max\bigg\{\mathbb{E}_{\mathcal{T}}\Big(\max_{i,j}\sum_{k=1}^{n}\mathcal{T}_{ijk}^2\Big)^{1/2}, \mathbb{E}_{\mathcal{T}}\Big(\max_{i,k}\sum_{j=1}^{n}\mathcal{T}_{ijk}^2\Big)^{1/2}, \mathbb{E}_{\mathcal{T}}\Big(\max_{j,k}\sum_{i=1}^{n}\mathcal{T}_{ijk}^2\Big)^{1/2}\bigg\},$$

$$\leq \mathbb{E}_{\mathcal{T}}\Big(\max_{i,j}\sum_{k=1}^{n}\mathcal{T}_{ijk}^2\Big)^{1/2} + \mathbb{E}_{\mathcal{T}}\Big(\max_{i,k}\sum_{j=1}^{n}\mathcal{T}_{ijk}^2\Big)^{1/2} + \mathbb{E}_{\mathcal{T}}\Big(\max_{j,k}\sum_{i=1}^{n}\mathcal{T}_{ijk}^2\Big)^{1/2}\bigg\},$$

$$\leq K\sqrt{n}$$

and

$$\mathbb{E}_{\mathcal{U}}\alpha_2 = \max\bigg\{\mathbb{E}_{\mathcal{U}}\Big(\max_{i,j}\sum_{k=1}^{n}\mathcal{U}_{ijk}^2\Big)^{1/2}, \mathbb{E}_{\mathcal{U}}\Big(\max_{i,k}\sum_{j=1}^{n}\mathcal{U}_{ijk}^2\Big)^{1/2}, \mathbb{E}_{\mathcal{U}}\Big(\max_{j,k}\sum_{i=1}^{n}\mathcal{U}_{ijk}^2\Big)^{1/2}\bigg\},$$

$$\leq \mathbb{E}_{\mathcal{U}}\Big(\max_{i,j}\sum_{k=1}^{n}\mathcal{U}_{ijk}^2\Big)^{1/2} + \mathbb{E}_{\mathcal{U}}\Big(\max_{i,k}\sum_{j=1}^{n}\mathcal{U}_{ijk}^2\Big)^{1/2} + \mathbb{E}_{\mathcal{U}}\Big(\max_{j,k}\sum_{i=1}^{n}\mathcal{U}_{ijk}^2\Big)^{1/2}\bigg\},$$

$$\leq K'\sqrt{n}.$$

Now, we can update (14) as

$$\mathbb{E}a\|\mathcal{T} \times_1 x \times_2 y\|_2 + b\|\mathcal{U} \times_1 x \times_2 z\|_2$$

$$\leq c_1\sqrt{2\pi}\left[a2^{3K+K'}K\sqrt{n}\big(\log_2(1/\eta)\big)^{K-1} + b2^{K+3K'}K'\sqrt{n}\big(\log_2(1/\eta)\big)^{K'-1}\right.$$

$$\left. + a2^{K-1}\sqrt{\eta n} + b2^{K'-1}\sqrt{\eta n}\right]\sqrt{2(K+K')\ln(5e/\eta)}.$$

By selecting $\eta = (\ln n)^{2(\max(K,K')-1)}/n$, we have

$$\mathbb{E}a\|\mathcal{T} \times_1 x \times_2 y\|_2 + b\|\mathcal{U} \times_1 x \times_2 z\|_2$$

$$\leq c_3\left[a2^{3K+K'}K\sqrt{n}(\ln n)^{K-1/2} + b2^{3K+K'}K'\sqrt{n}(\ln n)^{K'-1/2}\right].$$

$\square$

## C.2 Excess Risk Bounds

Next three theorems give excess risk bounds for coupled nuclear norms introduced in Section 4.

*Proof of Theorem 1*: We can expand (8) as

$$R_{\text{S,P}}(l \circ \mathcal{W}, l \circ \mathcal{V}) = \frac{1}{|\text{S} \cup \text{P}|}\mathbb{E}_\sigma\left[\sup_{\mathcal{W},\mathcal{V}\in\text{W}} \sum_{i_1,..,i_K} \Sigma_{i_1,...,i_K} l(\mathcal{X}_{i_1,...,i_K}, \mathcal{W}_{i_1,...,i_K})\right.$$

$$\left. + \sum_{j_1,...,j_{K'}} \Sigma'_{j_1,...,j_{K'}} l(\mathcal{Y}_{j_1,...,j_{K'}}, \mathcal{V}_{j_1,...,j_{K'}})\right]$$

$$\leq \frac{\Lambda}{|\text{S} \cup \text{P}|}\mathbb{E}_\sigma\left[\sup_{\mathcal{W},\mathcal{V}\in\text{W}} \sum_{i_1,..,i_K} \Sigma_{i_1,...,i_K} \mathcal{W}_{i_1,..,i_K}\right.$$

$$\left. + \sum_{j_1,...,j_{K'}} \Sigma'_{j_1,...,j_{K'}} \mathcal{V}_{j_1,...,j_{K'}}\right] \text{ (Lipschitz continuity)}$$

$$\leq \frac{\Lambda}{|\text{S} \cup \text{P}|}\mathbb{E}_\sigma\left[\sup_{\mathcal{W},\mathcal{V}\in\text{W}} \|\mathcal{W}\|_\star\|\Sigma\|_2 + \|V\|_\star\|\Sigma'\|_2\right], \text{ (Duality relationship)}$$

where the second line is by using the contraction inequality due to Lipschitz continuity of the loss function $l(\cdot, \cdot)$ and the last bound is obtained by applying the Holder's inequality. We want to point out that since $\mathcal{W}$ and $\mathcal{V}$ belongs to the hypothesis class W and they are constrained by the $\|\mathcal{W}, \mathcal{V}\|^a_{\text{ccp},(\lambda_1,\text{F})(\lambda_2,\text{F})}$. This indicates that $\|\Sigma\|_2$ and $\|\Sigma'\|_2$ are constrained by the dual norm $\|\Sigma, \Sigma'\|^a_{\text{ccp},(\lambda_1,\text{F})(\lambda_2,\text{F})^\star}$.

By the definition of the coupled norm (2), both $\mathcal{W}$ and $\mathcal{V}$ have rank $r$. Further, by taking upper bounds on $\gamma_1$ and $\mu_1$ as $\gamma_1 \leq B_\mathcal{W}$ and $\mu_1 \leq B_\mathcal{V}$, respectively, we obtain

$$\|\mathcal{W}\|_* \leq rB_\mathcal{W},$$

and

$$\|\mathcal{V}\|_* \leq rB_\mathcal{V}.$$

Then we can bound the Rademacher complexity as

$$R_{\text{S,P}}(l \circ \mathcal{W}, l \circ \mathcal{V}) \leq \frac{\Lambda}{|\text{S} \cup \text{P}|}\mathbb{E}_\sigma\left[r\gamma_1\|\Sigma\|_2 + r\mu_1\|\Sigma'\|_2\right].$$

Using Theorem 5 with $a = r\gamma_1$ and $b = r\mu_1$, we obtain the desired bound

$$R_{\text{S,P}}(l \circ \mathcal{W}, l \circ \mathcal{V}) \leq \frac{c\Lambda}{|\text{S} \cup \text{P}|}\left[rB_\mathcal{W}2^{3K+K'}K\sqrt{n}(\ln n)^{K-1/2}\right.$$

$$\left. + rB_\mathcal{V}2^{K+3K'}K'\sqrt{n}(\ln n)^{K'-1/2}\right].$$

$\square$

*Proof of Theorem 2*: We expand (8) for the coupled norm $\|\mathcal{W}, \mathcal{V}\|^a_{\mathrm{ccp},(\lambda_1,\lambda_2,\mathrm{L})(\lambda_3,\mathrm{F})}$ as

$$
\begin{aligned}
R_{\mathrm{S,P}}(l \circ \mathcal{W}, l \circ \mathcal{V}) &= \frac{1}{|\mathrm{S} \cup \mathrm{P}|}\mathbb{E}_\sigma \Bigg[ \sup_{\mathcal{W}^{(1)}, \mathcal{W}^{(2)}, \mathcal{V} \in \mathsf{W}} \sum_{i_1,..,i_K} \Sigma_{i_1,..,i_K} l(\mathcal{X}_{i_1,..,i_K}, \mathcal{W}^{(1)}_{i_1,..,i_K} + \mathcal{W}^{(2)}_{i_1,..,i_K}) \\
&\qquad\qquad + \sum_{j_1,...,j_{K'}} \Sigma'_{j_1,...,j_{K'}} l(\mathcal{Y}_{j_1,...,j_{K'}}, \mathcal{V}_{j_1,...,j_{K'}}) \Bigg] \\
&\leq \frac{\Lambda}{|\mathrm{S} \cup \mathrm{P}|}\mathbb{E}_\sigma \Bigg[ \sup_{\mathcal{W}^{(1)}, \mathcal{W}^{(2)}, \mathcal{V} \in \mathsf{W}} \sum_{i_1,..,i_K} \Sigma_{i_1,..,i_K} (\mathcal{W}^{(1)}_{i_1,..,i_K} + \mathcal{W}^{(2)}_{i_1,..,i_K}) \\
&\qquad\qquad + \sum_{j_1,...,j_{K'}} \Sigma'_{j_1,...,j_{K'}} \mathcal{V}_{j_1,...,j_{K'}} \Bigg] \text{(Lipschitz continuity)} \\
&\leq \frac{\Lambda}{|\mathrm{S} \cup \mathrm{P}|}\mathbb{E}_\sigma \Bigg[ \sup_{\mathcal{W}^{(1)}, \mathcal{W}^{(2)}, \mathcal{V} \in \mathsf{W}} \|\mathcal{W}^{(1)}\|_\star \|\Sigma\|_{\star^*} + \|W^{(2)}\|_\star \|\Sigma\|_2 + \|\mathcal{V}\|_\star \|\Sigma'\|_2 \Bigg],
\end{aligned}
$$
$$\text{(Duality relationship)}$$

where in the last step we apply the Holder's inequality to each $\mathcal{W}^{(1)}$, $\mathcal{W}^{(2)}$, and $\mathcal{V}$ with relation to $\Sigma$ and $\Sigma'$. Since we use the hypothesis class $\mathsf{W}$ both $\Sigma$ and $\Sigma'$ are also constrained by the dual norm $\|\Sigma, \Sigma'\|^a_{\mathrm{ccp},(\lambda_1,\lambda_2,\mathrm{L})(\lambda_3,\mathrm{F})^*}$.

We introduce upper bounds for $\gamma_1^{(1)}$, $\gamma_1^{(2)}$, and $\mu_1$ of the coupled norm (3) as $B_{\mathcal{W}_1}$, $B_{\mathcal{W}_2}$, and $B_{\mathcal{V}}$, respectively. Since we take $\mathrm{Rank}(\mathcal{W}^{(1)}) = \mathrm{Rank}(\mathcal{V}) = r_1$ and by assumptions that $\gamma_1^{(1)} \leq B_{\mathcal{W}_1}$ and $\mu_1 \leq B_{\mathcal{V}}$, we obtain $\|\mathcal{W}^{(1)}\|_\star \leq r_1 B_{\mathcal{W}_1}$ and $\|\mathcal{V}\|_\star \leq r_1 B_{\mathcal{V}}$. Further, by assumption that $\gamma_1^{(2)} \leq B_{\mathcal{W}_2}$, we obtain $\|\mathcal{W}^{(2)}\|_\star \leq r_1 B_{\mathcal{W}_2}$. Then, using the definition of the hypothesis class, we have

$$
R_{\mathrm{S,P}}(l \circ \mathcal{W}, l \circ \mathcal{V}) \leq \frac{c\Lambda}{|\mathrm{S} \cup \mathrm{P}|}\mathbb{E}_\sigma \Big[ r_1 B_{\mathcal{W}_1} \|\Sigma\|_2 + r_2 B_{\mathcal{W}_2} \|\Sigma\|_2 + r_2 B_{\mathcal{V}} \|\Sigma'\|_2 \Big].
$$

Using the Theorem 5, we obtain

$$
\begin{aligned}
R_{\mathrm{S,P}}(l \circ \mathcal{W}, l \circ \mathcal{V}) \leq \frac{c\Lambda}{|\mathrm{S} \cup \mathrm{P}|}\Bigg[ &(r_1 B_{\mathcal{W}_1} + r_2 B_{\mathcal{W}_2})2^{3K+K'} K\sqrt{n}(\ln n)^{K-1/2} \\
&+ r_2 B_{\mathcal{V}} 2^{K+3K'} K'\sqrt{n}(\ln n)^{K'-1/2} \Bigg].
\end{aligned}
$$

$\square$

Additionally, we give the excess risk bound for completion using the norm $\|\mathcal{W}, \mathcal{V}\|^a_{\mathrm{ccp},(\lambda_1,\lambda_2,\mathrm{L})(\lambda_3,\lambda_4,\mathrm{L})}$ in the next theorem.

**Theorem 6.** *Let us consider* $\|\mathcal{W}, \mathcal{V}\|^a_{\mathrm{ccp},(\lambda_1,\lambda_2,\mathrm{L})(\lambda_3,\lambda_4,\mathrm{L})}$ *and its hypothesis class as* $\mathsf{W} = \{\mathcal{W}^{(1)}, \mathcal{W}^{(2)}, \mathcal{V}^{(1)}, \mathcal{V}^{(2)}$ : $\inf_{\mathcal{W}^{(1)}+\mathcal{W}^{(2)}=\mathcal{W}} \inf_{\mathcal{V}^{(1)}+\mathcal{V}^{(2)}=\mathcal{V}} \|\mathcal{W}, \mathcal{V}\|^a_{\mathrm{ccp},(\lambda_1,\lambda_2,\mathrm{L})(\lambda_3,\lambda_4,\mathrm{L})}, \mathrm{rank}(\mathcal{W}^{(1)}) = \mathrm{rank}(\mathcal{V}^{(1)}) = r_1, \mathrm{rank}(\mathcal{W}^{(2)}) = r_2, \mathrm{rank}(\mathcal{V}^{(2)}) = r_3\}$, *the Rademacher complexity is bounded as*

$$
\begin{aligned}
R_{\mathrm{S,P}}(l \circ \mathcal{W}, l \circ \mathcal{V}) \leq \frac{c\Lambda}{|\mathrm{S} \cup \mathrm{P}|}\Bigg[ &(r_1 B_{\mathcal{W}_1} + r_2 B_{\mathcal{W}_2})2^{3K+K'} K\sqrt{n}(\ln n)^{K-1/2} \\
&+ (r_1 B_{\mathcal{V}_1} + r_3 B_{\mathcal{V}_2})2^{K+3K'} K'\sqrt{n}(\ln n)^{K'-1/2} \Bigg].
\end{aligned}
$$

*where* $\gamma_1^{(1)} \leq B_{\mathcal{W}_1}$, $\gamma_1^{(2)} \leq B_{\mathcal{W}_2}$, $\nu_1^{(1)} \leq B_{\mathcal{V}_1}$, $\nu_1^{(2)} \leq B_{\mathcal{V}_2}$, *and c is a constant.*

*Proof*: We can expand (8) for $\|\mathcal{W}, \mathcal{V}\|_{\text{ccp},(\lambda_1,\lambda_2,\text{L})(\lambda_3,\lambda_4,\text{L})}^a$ as

$$R_{\text{S,P}}(l \circ \mathcal{W}, l \circ \mathcal{V}) = \frac{1}{|\text{S} \cup \text{P}|} \mathbb{E}_\sigma \Bigg[$$

$$\sup_{\mathcal{W}^{(1)},\mathcal{W}^{(2)},\mathcal{V}^{(1)},\mathcal{V}^{(2)} \in \mathsf{W}} \sum_{i_1,..,i_K} \Sigma_{i_1,..,i_K} l(\mathcal{T}_{i_1,..,i_K}, \mathcal{W}^{(1)}_{i_1,..,i_K} + \mathcal{W}^{(2)}_{i_1,..,i_K})$$

$$+ \sum_{j_1,...,j_{K'}} \Sigma'_{j_1,...,j_{K'}} l(\mathcal{Y}_{j_1,...,j_{K'}}, \mathcal{V}^{(1)}_{j_1,...,j_{K'}} + \mathcal{V}^{(2)}_{j_1,...,j_{K'}}) \Bigg]$$

$$\leq \frac{\Lambda}{|\text{S} \cup \text{P}|} \mathbb{E}_\sigma \Bigg[ \sup_{\mathcal{W}^{(1)},\mathcal{W}^{(2)},\mathcal{V}^{(1)},\mathcal{V}^{(2)} \in \mathsf{W}} \sum_{i_1,..,i_K} \Sigma_{i_1,..,i_K} (\mathcal{W}^{(1)}_{i_1,..,i_K} + \mathcal{W}^{(2)}_{i_1,..,i_K})$$

$$+ \sum_{j_1,...,j_{K'}} \Sigma'_{j_1,...,j_{K'}} (\mathcal{V}^{(1)}_{j_1,...,j_{K'}} + \mathcal{V}^{(2)}_{j_1,...,j_{K'}}) \Bigg] \quad \text{(Lipschitz continuity)}$$

$$\leq \frac{\Lambda}{|\text{S} \cup \text{P}|} \mathbb{E}_\sigma \Bigg[ \sup_{\mathcal{W}^{(1)},\mathcal{W}^{(2)},V^{(1)},V^{(2)} \in \mathsf{W}} \|\mathcal{W}^{(1)}\|_\star \|\Sigma\|_{\star^*} + \|\mathcal{W}^{(2)}\|_\star \|\Sigma\|_2$$

$$+ \|\mathcal{V}^{(1)}\|_\star \|\Sigma'\|_2 + \|\mathcal{V}^{(2)}\|_\star \|\Sigma'\|_2 \Bigg], \quad \text{(Duality relationship)}$$

where in the last step we apply the Holder's inequality to each $\mathcal{W}^{(1)}$, $\mathcal{W}^{(2)}$, $\mathcal{V}^{(1)}$, and $\mathcal{V}^{(2)}$ with relation to $\Sigma$ and $\Sigma'$. Since we use the hypothesis class $\mathsf{W}$ both $\Sigma$ and $\Sigma'$ are also constrained by the dual norm $\|\Sigma, \Sigma'\|_{\text{ccp},(\lambda_1,\lambda_2,\text{L})(\lambda_3,\lambda_4,\text{L})}^a$.

Now, we use upper bounds for $\gamma_1^{(1)}$, $\gamma_1^{(2)}$, $\mu_1^{(1)}$, and $\mu_1^{(2)}$ of the coupled norm (3) as $B_{\mathcal{W}_1}$, $B_{\mathcal{W}_2}$, $B_{\mathcal{V}_1}$, and $B_{\mathcal{V}_2}$, respectively. Form the definition of the norm, we know that $\text{Rank}(\mathcal{W}^{(1)}) = \text{Rank}(\mathcal{V}^{(1)}) = r_1$, $\text{Rank}(\mathcal{W}^{(2)}) = r_2$, and $\text{Rank}(\mathcal{V}^{(2)}) = r_3$. By assumptions that $\gamma_1^{(1)} \leq B_{\mathcal{W}_1}$ and $\mu_1^{(1)} \leq B_{\mathcal{V}_1}$, we obtain $\|\mathcal{W}^{(1)}\|_\star \leq r_1 B_{\mathcal{W}_1}$, $\|\mathcal{W}^{(2)}\|_\star \leq r_2 B_{\mathcal{W}_2}$, $\|\mathcal{V}^{(1)}\|_\star \leq r_1 B_{\mathcal{V}_1}$, and $\|\mathcal{V}^{(2)}\|_\star \leq r_3 B_{\mathcal{V}_2}$. These assumptions lead to

$$R_{\text{S,P}}(l \circ \mathcal{W}, l \circ \mathcal{V}) \leq \frac{\Lambda}{|\text{S} \cup \text{P}|} \mathbb{E}_\sigma \Big[ r_1 B_{\mathcal{W}_1} \|\Sigma\|_2 + r_2 B_{\mathcal{W}_2} \|\Sigma\|_2 + r_1 B_{\mathcal{V}_1} \|\Sigma'\|_2 + r_3 B_{\mathcal{V}_2} \|\Sigma'\|_2 \Big] .$$

Using the Theorem 5, we obtain

$$R_{\text{S,P}}(l \circ \mathcal{W}, l \circ \mathcal{V}) \leq \frac{c\Lambda}{|\text{S} \cup \text{P}|} \Bigg[ (r_1 B_{\mathcal{W}_1} + r_2 B_{\mathcal{W}_2}) 2^{3K+K'} K \sqrt{n} (\ln n)^{K-1/2}$$

$$+ (r_1 B_{\mathcal{V}_1} + r_3 B_{\mathcal{V}_2}) 2^{K+3K'} K' \sqrt{n} (\ln n)^{K'-1/2} \Bigg].$$

$\square$

As a reference, we give the excess risk bounds for tensor completion that is regularized using the tensor nuclear norm. In order to prove this, we use the following theory from (Nguyen et al., 2015).

**Theorem 7.** *Let $\mathcal{T} \in \mathbb{R}^{n \times \cdots \times n}$ be a random $K$-mode tensor, whose entries are independent, zero-mean random variables. For any $\kappa \leq \frac{1}{64}$, assume that $1 \leq q \leq 2K\kappa n \ln \frac{5e}{\kappa}$. Then,*

$$(\mathbb{E}\|\mathcal{T}\|_2^q)^{\frac{1}{q}} \leq c' 8^K \sqrt{2K \ln \frac{5e}{\kappa}} \left( \left[ \log_2 \left( \frac{1}{\kappa} \right) \right]^{K-1} \left( \sum_{j=1}^K \mathbb{E}_\mathcal{T} \alpha_j^q \right)^{\frac{1}{q}} + \sqrt{\kappa n} (\mathbb{E}_\mathcal{T} \beta^q)^{\frac{1}{q}} \right),$$

*where*

$$\alpha_j^2 = \max_{i_1,\ldots,i_{j-1},i_{j+1},\ldots,i_K} \left( \sum_{i_j}^n \mathcal{T}_{i_1,\ldots,i_{j-1},i_{j+1},\ldots,i_K}^2 \right) \quad \text{and} \quad \beta = \max_{i_1,\ldots,i_K} |\mathcal{T}_{i_1,\ldots,i_K}|$$

Since we use the spectral norm of a tensor, $q = 1$. Additionally, we want to consider the case where $\mathcal{T}_{i_1,\dots,i_K} \in \{1, 0, -1\}$ to bounds Rademacher variables in our proofs.

With $q = 1$, we have

$$\mathbb{E}\|\mathcal{T}\|_2 \leq c'8^K \sqrt{2K\ln\frac{5e}{\kappa}}\left(\left[\log_2\left(\frac{1}{\kappa}\right)\right]^{K-1}\left(\sum_{j=1}^{K}\mathbb{E}_{\mathcal{T}}\alpha_j\right) + \sqrt{\kappa n}\right),$$

where

$$\alpha_j^2 = \max_{i_1,\dots,i_{j-1},i_{j+1},\dots,i_K}\left(\sum_{i_j}^{n}\mathcal{T}_{i_1,\dots,i_{j-1},i_{j+1},\dots,i_K}^2\right) \quad \text{and} \quad \beta = \max_{i_1,\dots,i_K}|\mathcal{T}_{i_1,\dots,i_K}|.$$

Since $\mathcal{T}_{i_1,\dots,i_K} \in \{1, 0, -1\}$ we have $\beta = 1$ and $\mathbb{E}_{\mathcal{T}}\alpha_j \leq \sqrt{n}$ leading to

$$\mathbb{E}\|\mathcal{T}\|_2 \leq c'8^K\sqrt{2K\ln\left(\frac{5e}{\kappa}\right)}\left(\left[\log_2\left(\frac{1}{\kappa}\right)\right]^{K-1}K\sqrt{n} + \sqrt{\kappa n}\right).$$

Let us consider $\kappa = \frac{(\ln n)^{2(K-1)}}{n}$ as in the Corollary 4 in (Nguyen et al., 2015), then we obtain

$$\mathbb{E}\|\mathcal{B}\|_2 \leq c'8^K K(\ln n)^{K-1/2}\sqrt{n}. \tag{34}$$

In the theorem below, we give the excess risk bound for the individual completion $\mathcal{T}$ using the tensor nuclear norm (1).

**Theorem 8.** *Let us consider a $K$-mode tensor $\mathcal{W} \in \mathbb{R}^{n \times \cdots \times n}$ with observed samples indexed by the set $S$. Let the hypothesis class for completion of $\mathcal{T}$ using the tensor nuclear norm be $\mathsf{W} = \{\mathcal{W} \mid \|\mathcal{W}\|_* \leq B, \ \text{rank}(\mathcal{W}) = r\}$. Then following Rademacher complexity holds with probability $1 - \delta$,*

$$R_S(l \circ \mathcal{W}) \leq \frac{\Lambda}{|S|}c'8^K r B_{\mathcal{W}} K(\ln n)^{K-1/2}\sqrt{n}.$$

*where $\gamma_1 \leq B_{\mathcal{W}}$ of (1) and $c'$ is a constant.*

*Proof*: The Rademacher complexity for a individual tensor can be written as follows,

$$R_S(l \circ \mathcal{W}) = \frac{1}{|S|}\mathbb{E}_\sigma\left[\sup_{\mathcal{W} \in \mathsf{W}}\sum_{(i_1,\dots,i_K)\in S}\Sigma_{i_1,\dots,i_K}l(\mathcal{W}_{i_1,\dots,i_K}, \mathcal{T}_{i_1,\dots,i_K})\right],$$

where $\Sigma_{i_1,\dots,i_K} = \sigma_\phi \in \{-1, 1\}$ with probability $0.5$ if $(i_1,\dots,i_K) \in S$ belonging to an index $\phi \in 1,\dots,|S|$ or $\Sigma_{i_1,\dots,i_K} = 0$ otherwise, which can be further expanded as

$$R_S(l \circ \mathcal{W}) = \frac{\Lambda}{|S|}\mathbb{E}_\sigma\left[\sup_{\mathcal{W} \in \mathsf{W}}\sum_{i_1,\dots,i_K}\Sigma_{i_1,\dots,i_K}l(\mathcal{W}_{i_1,\dots,i_K}, \mathcal{T}_{i_1,\dots,i_K})\right]$$

$$\leq \frac{\Lambda}{|S|}\mathbb{E}_\sigma\left[\sup_{\mathcal{W} \in \mathsf{W}}\sum_{i_1,\dots,i_K}\Sigma_{i_1,\dots,i_K}\mathcal{W}_{i_1,\dots,i_K}\right] \quad \text{(Lipschitz continuity)}$$

$$\leq \frac{\Lambda}{|S|}\mathbb{E}_\sigma\left[\sup_{\mathcal{W} \in \mathsf{W}}\|\mathcal{W}\|_\star\|\Sigma\|_2\right] \quad \text{(Duality relationship)}.$$

By the definition of the tensor nuclear norm, we have

$$\|\mathcal{W}\|_* = \inf\left\{\sum_{j=1}^{r}\gamma_j \Big| \mathcal{W} = \sum_{j=1}^{r}\gamma_j u_{1j} \otimes u_{2j} \otimes \cdots \otimes u_{Kj}, \|u_{kj}\|_2^2 = 1, \gamma_j \geq \gamma_{j+1} > 0\right\},$$

and with an assumption that $\gamma_1 \leq B_{\mathcal{W}}$, we have $\|\mathcal{W}\|_\star \leq r B_{\mathcal{W}}$. Further, using the result in (34), we obtain the desired bound

$$R_S(l \circ \mathcal{W}) \leq \frac{\Lambda}{|S|}c'8^K r B_{\mathcal{W}} K(\ln n)^{K-1/2}\sqrt{n}.$$

$\square$

# D More Simulation Experiments

In this section, we provide more simulation experiments for coupled tensors based on different ranks as we have given in the Section 5.1. We consider the same conditions and coupled tensor structures $\mathcal{T} \in \mathbb{R}^{20 \times 20 \times 20}$ and $M \in \mathbb{R}^{20 \times 30}$.

Our first experiment in this section was designed by making the tensor $\mathcal{T}$ with CP rank of $5$ and the matrix $M$ with rank of $10$. We shared $5$ components on the first modes to couple $\mathcal{T}$ and $M$. Figure 4 shows that coupled norms have given an equivalent performance to the tensor nuclear norm for tensor completion. For matrix completion, we can see that the best performance is given by $\|\cdot, \cdot\|_{\text{ccp},(\lambda,\lambda,\text{L}),(\lambda,\text{F})}$ (ccp-3), better than the other two coupled nuclear norms. This indicates that $\|\cdot, \cdot\|_{\text{ccp},(\lambda,\lambda,\text{L}),(\lambda,\text{F})}$ is able to learn more efficiently since it separates the shared and unshared components among the coupled tensor and the matrix.

(a) Matrix Completion ($M$)     (b) Tensor Completion ($\mathcal{T}$)

Figure 4: Performances of completion of the tensor with dimensions of $20 \times 20 \times 20$ and CP rank of $5$ and matrix with dimensions of $20 \times 30$ and rank of $10$ both sharing $5$ components.

Next, we designed a tensor $\mathcal{T}$ with multilinear rank of $(15, 5, 5)$ and a matrix $M$ with rank of $5$. Again, we shared $5$ components on the first modes to couple $\mathcal{T}$ and $M$. Figure 5 shows that for tensor completion, coupled nuclear norms have given a comparable performance to tensor nuclear norm and the coupled norm $(\text{O}, \text{O}, \text{S})$. However, for matrix completion coupled nuclear norms have dominated when the training samples are small and have given a weaker performance as the number of training samples increases.

(a) Matrix Completion ($M$)     (b) Tensor Completion ($\mathcal{T}$)

Figure 5: Performances of completion of the tensor with dimensions of $20 \times 20 \times 20$ and multilinear rank of $(15, 5, 5)$ and matrix with dimensions of $20 \times 30$ and rank of $5$ both sharing $5$ components.