[Reviews · NeurIPS 2018]

Reviewer 1



Two tensors are considered to be coupled when they share a common mode. This paper studies the convex coupled norms for coupled tensor completion. Several coupled tensor completion models have been proposed but many of them are nonconvex which leads to local optimal solutions. This work instead focused the study the convex norms. The main contribution is to propose a set of coupled norms extending the tensor nuclear norm that are convex, leading to global optimal solutions. The authors then present the theoretical analysis for tensor completion based on the proposed tensor nuclear norm. Generally, this work is novel. Looking for the convex relaxation of tensor ranks is always important. Giving the theoretical bound for tensor completion is important. The experiments verify the theoretical bound. I have a question about the achieved bound in Section 4. Is the obtained bound optimal? Does there exist the optimal bound? In the experiments, the authors compared the proposed tensor completion model with some existing ones. The authors may miss some other related work about tensor completion based on different tensor ranks and tensor nuclear norm, e.g., Ji Liu, Przemyslaw Musialski, Peter Wonka, and Jieping Ye. Tensor completion for estimating missing values in visual data. TPAMI, 35(1):208–220, 2013. Cun Mu, Bo Huang, John Wright, and Donald Goldfarb. Square deal: Lower bounds and improved relaxations for tensor recovery. In ICML, pages 73–81, 2014. Canyi Lu, Jiashi Feng, Zhouchen Lin and Shuicheng Yan, Exact Low Tubal Rank Tensor Recovery from Gaussian Measurements, IJCAI, 2018

Reviewer 2



PAPER SUMMARY This paper considers the problem of completion of coupled tensors. The main contribution of the paper is the proposal of coupled norms of coupled tensors. These norms are based on the tensor nuclear norm and as functions of the tensors they are convex, thus allowing for globally optimal solutions, once incorporated into optimization problems for tensor completion. The paper also establishes that the excess risk bounds associated with the proposed norms are smaller than the bounds associated to norms that penalize the multilinear rank of the coupled tensors. Experiments on synthetic data and on a real location-user-activity dataset (UCLAF) show advantages of the proposed completion models to existing ones. COMMENTS My major comment is to put some more effort into clarifying the presentation for the sake of the non-expert reader. At several places in the paper the notation is unclear or motivation is lacking. 1. Some motivation and review of coupled tensors is missing. 2. Some minor issues in the use of English need to be corrected, especially articles are often missing. 3. Please define [M;N] mathematically. 4. Equation between lines 98-99: a comment about how this definition reduces to the nuclear norm for the matrix case is in order. 5. I don't understand eq. (1): What is X and Y? Also, what is the value of the norm? Not, clearly defined. 6. Equation (2): Margin violated. Again, i don't understand how this norm is defined. Is it the infimum of ||W^(1)||*, ||W^(2)||*, ||V||*? What if we take W^(1)=0? What if the constraints being less than \lambda_1, \lambda_2, \lambda_3 are not satisfied at the same time?

Reviewer 3



The paper studies the problem of coupled tensor completion, which is originally motivated from multi-task settings. The proposed method, coupled nuclear norm, extends the previously proposed tensor nuclear norm to the setting where two tensors coupled in some mode-k. Such an extension is easy to think of and does not need a lot of effort. The authors show that, using the proposed coupled nuclear norm, the risk bound is reduced, saying from O(n^0.5(K-1)) to O(n^0.5 log(n)^K); this is a distinct improvement (but the criterion used to measure the risk is questionable). Comments: 1. The proposed method as well as analysis techniques are some kind of mixture of existing papers. Thus, the innovation level of this paper is not high. But this is okay to me, as long as the authors can make a solid contribution to the community. 2. The major concern is about the clarify, in particular the confusing notations used throughout the paper. a) As the tensors are coupled in only one mode, it is necessary to use a superscript a to specify the coupled mode. Also, it is not a good idea to begin the method with some complicated notation with subscripts (b,c,d). I think it is beneficial to keep the notations as simple as possible. b) The proposed coupled nuclear norm defined in Equation (1) seems a set, not a norm. In particular, what are \mathcal{X} and \mathcal{Y}? The are similar problems with Equation (2). c) On the theoretical analysis, it is really wired to split the observed entries into a training set and a test set. Usually, the test data should be the missing entries. d) The Rademacher complexity measures only the richness of the proposed norm. However, in general, more freedoms do not necessarily lead to good recovery performance. It will be more convincing if the authors can prove that the proposed coupled nuclear norm can reduce the sampling complexity required to restore the missing entries.